

# An automatic mesh generator for coupled 1D/2D hydrodynamic models

Younghun Kang[1] and Ethan J. Kubatko[1]

[1]Department of Civil, Environmental, and Geodetic Engineering, The Ohio State University, Columbus, Ohio, USA, 43210

**Correspondence:** Younghun Kang (kang.1049@buckeyemail.osu.edu)

**Abstract.** Two-dimensional (2D), depth-averaged shallow water equation (SWE) models are routinely used to simulate flooding in coastal areas — areas that often include vast networks of channels and flood-control topographic features/structures, such as barrier islands and levees. Adequately resolving these features within the confines of a 2D model can be computationally expensive, which has led to coupling 2D simulation tools to less-expensive, one-dimensional (1D) models. Under certain 1D/2D coupling approaches, this introduces internal constraints that must be considered in the generation of the 2D computational mesh used. In this paper, we further develop an existing automatic unstructured mesh generation tool for SWE models, ADMESH+, to sequentially (i) identify 1D constraints from the raw input data used in the mesh generation process, namely, the digital elevation model (DEM) and land/water delineation data, (ii) distribute grid points along these internal constraints according to feature curvature and user-prescribed minimum grid spacing, and (iii) integrate these internal constraints into the 2D mesh-size-function and mesh-generation processes. The developed techniques, which include a novel approach for determining the so-called medial axis of a polygon, are described in detail and demonstrated on three test cases, including two inland watersheds with vast networks of channels and a complex estuarian system on the Texas, US coast.

## 1 Introduction

Hydrodynamic models are routinely used as to simulate, analyze, and assess the effects of physical phenomenon that result in coastal flooding, such as sea-level rise and hurricane storm surge. Typically, the two-dimensional (2D), depth-averaged shallow water equations, equipped with a suitable wetting and drying algorithm, are used to model inundation within the coastal floodplain in the main (see, for example, Luettich and Westerink (1999); Bunya et al. (2009); Dawson et al. (2011)). However, these coastal regions often include vast networks of small-scale channels that, while playing a significant part in the conveyance of flood waters propagating into and through the floodplain, are often left under-resolved in practice due to the computational expensive of adequately resolving them within the confines of a strictly 2D modeling approach. Moreover, in many situations, flow in open-channels (and storm sewer systems in urban areas) can be adequately described by simpler one-dimensional (1D), section-averaged flow equations. This situation has led to the development of a number of coupled 1D/2D modeling approaches over the past decade (see references below) that aim to more accurately and efficiently simulate flooding events, and coastal hydrodynamics in general, than traditional 2D modeling approaches.



These types of coupled models have been applied in a number of different hydrodynamic scenarios but have been most widely used to simulate river flooding problems in various settings (e.g., Liu et al. (2015); D'Alpaos and Defina (2007); Kuiry et al. (2010); Martini et al. (2004); Marin and Monnier (2009); Gejadze and Monnier (2007); Timbadiya et al. (2015); Stelling and Verwey (2006b); Li et al. (2021); Morales-Hernández et al. (2016)). Some recent investigations have focused on inundation in urban areas where storm sewer systems and channels, which are approximated in 1D, interact with a 2D flooding model

(Vojinovic and Tutulic (2009); Adeogun et al. (2012, 2015); Delelegn et al. (2011); Seyoum et al. (2012)). Other coupled models have been developed and applied for coupled riverine-estuarine flows near coastal areas (Bakhtyar et al. (2020); Lin et al. (2006)), river-lake flows (Chen et al. (2012); Pham Van et al. (2016)), supercritical flow in crossroads (Ghostine et al. (2015)), overland-open-channel flows (West et al. (2017)), and river closure projects (Lin et al. (2020)). More examples of coupled models, including commercial models, are summarized in Néelz and Pender (2009); Teng et al. (2017); Woodhead

et al. (2007).

The coupled models are often categorized based on the types of interactions that occur between the 1D/2D flows, but here we categorize them into three types based on the way the 1D and 2D domains are connected (see Fig. 1). The first type is *boundary-connected* domains (e.g., Chen et al. (2012); Liu et al. (2015); Bakhtyar et al. (2020); Ghostine et al. (2015); Pham Van et al. (2016)). These types of domains are widely used for river-lake or river-estuary systems (Chen et al. (2012);

Bakhtyar et al. (2020); Pham Van et al. (2016); Ghostine et al. (2015)), where the longitudinal (or frontal) flows from 1D domains enter the 2D domains as a boundary condition. However, in some cases, the interaction can be made by lateral flow through breach between the 1D/2D domains (Liu et al. (2015)). The second type is *internally-connected* domains (e.g., West et al. (2017); Kuiry et al. (2010); D'Alpaos and Defina (2007); Martini et al. (2004); Marin and Monnier (2009); Gejadze and Monnier (2007); Stelling and Verwey (2006a); Timbadiya et al. (2015); Vojinovic and Tutulic (2009)). These types of domains

are widely used for river-floodplain systems. The interaction is made along the whole 1D domain, where the discharge from 2D domains can enter the 1D domains or the 1D channel flows exceeding bank level can enter the 2D domain. The third type is *vertically-connected* domains (e.g., Vojinovic and Tutulic (2009); Fan et al. (2017); Adeogun et al. (2012, 2015); Delelegn et al. (2011)). These types of domains are mostly used for urban inundation with storm sewer systems. The interaction is made at points where 1D/2D domains are connected vertically, where the surcharged overflow can enter the 2D domains.

Depending on the type of 1D/2D domain connections, the mesh generation can be straightforward or extremely complicated. The simplest case is *boundary-connected* domains. In this case the 2D computational meshes can be generated with automatic mesh generators that have been developed for 2D hydrodynamic models; see, for example, Persson and Strang (2004); Conroy et al. (2012); Koko (2015); Roberts et al. (2019); Engwirda (2017); Hagen et al. (2002); Bilgili et al. (2006); Candy and Pietrzak (2018); Avdis et al. (2018); Remacle and Lambrechts (2018); Gorman et al. (2006). It would then remain to generate

1D meshes and to connect them to the 2D mesh at the boundary, which is straightforward once the 1D domain is generated. For the *vertically-connected* domains, staggered 1D/2D computational meshes are widely used, i.e., 1D line elements are not constrained to be collocated with 2D mesh edges, given that the connections between the 1D and 2D domains are limited to points. For example, the meshes used in Delelegn et al. (2011); Adeogun et al. (2012, 2015) are generated with standard



two dimensional mesh generators, such as Gaja3D (Rath (2007)) and Triangle (Shewchuk (1996)), without consideration of
locations of links.

The most complicated case is the *internally-connected* domains. Unlike *vertically-connected* domains, collocated meshes, which here means that 1D line elements are aligned with 2D mesh edges, are desirable. The difficulty of mesh generation for this type comes from the fact that the resolutions of the 1D/2D domains are obviously closely intertwined with each other, however, the desired mesh resolutions for each domain may be quite different. Additionally, the 1D line elements must serve
as internal constraints in the mesh generation process, which must be carefully identified and pre-pocessed to avoid "over-constraining" the 2D elements, leading to triangles of poor quality.

In this paper, we present an automatic mesh generator for internally-connected 1D/2D hydrodynamic models that is extension of an existing mesh generator for SWE models, ADMESH+ (Conroy et al. (2012)), which built upon the ideas and methodology of Persson's DistMesh program — a simple, open-source mesh generator implemented in Matlab (Persson and Strang (2004)).
The ADMESH+ mesh generation process can be briefly outlined as follows. First, a mesh- or element-size function, $h$, is constructed that is used to prescribe element sizes $h(x)$ throughout a given 2D domain. These element sizes are based on a number of geometric factors, such as shoreline/boundary curvature and bathymetric/topographic gradients, as well as user-defined inputs, such as target maximum/minimum element sizes and mesh-grading specifications (i.e., the ratio of neighboring elements should not exceed some specified factor). Given the element-size function, a Delaunay triangulation of an initial set
of mesh nodes with a density proportional to $1/h(x)^2$ is then generated and the nodes of this initial mesh are re-positioned by solving for force equilibrium (iteratively) around each node, making use of a spring mechanics' analogy; see Conroy et al. (2012) for complete details.

The primary improvements incorporated into ADMESH+ in this work are twofold. First, automatic identification of 1D domains is developed. This requires as input to the mesh generation process a digital elevation model (DEM) and a so-called
land/water mask that identifies the (initially) "dry"/"wet" (respectively) portions of the domain. Using these input data sets, separate methods of identification of "narrow" channels (i.e., those below a user-specified 2D minimum elements size) that define 1D model domains over land and water are developed. These methods involve tightly integrating AMESH+ with a Matlab-based topographic toolbox, TopoToolbox (Schwanghart and Kuhn (2010)), and identifying the medial axis of the water portion of the domain, for which a novel approach is developed. Second, from the extracted 1D model domains, target 1D mesh
node distributions are computed along smooth spline approximations of the 1D line segments according to channel curvature, i.e., mesh node density is increased in areas of high curvature and relaxed in straight segments, as well as the underlying 2D mesh-size function computed during the standard mesh generation process. Similar to the 2D force-equilibrium approach mentioned above, the actual 1D mesh node distribution is then determined from the target mesh size through the use of a 1D spring mechanics' approach. The final internal constraints obtained are then used within the 2D mesh generation process to
obtain meshes suitable for coupled (specifically, internally-connected) 1D/2D hydrodynamic models.

The rest of this paper is organized as follows. In the next Section, the framework of the proposed methodology and the primary input data sets of the mesh generation are discussed. Details of the algorithms developed to identify internal constraints are then described in Section 3, with illustrative examples for simple and complex geometries. The 1D and 2D mesh generation





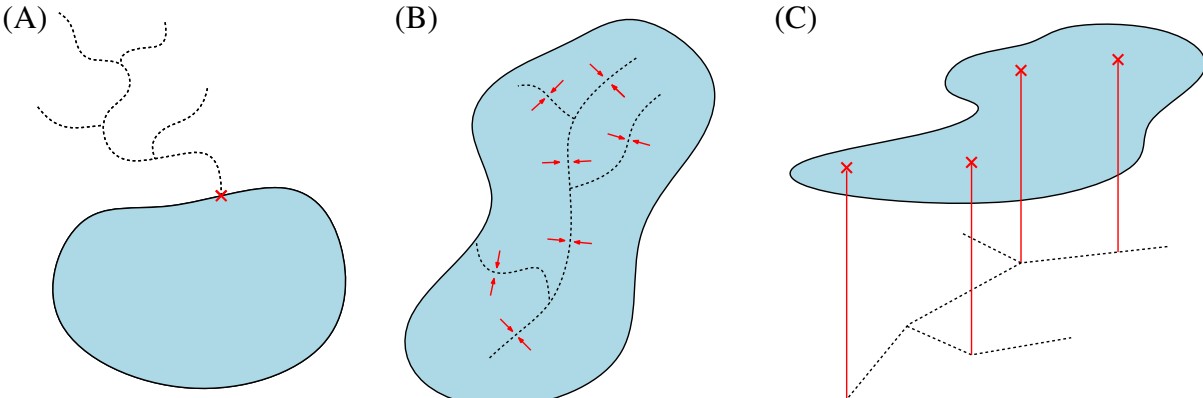

**Figure 1.** Schematic of three types of coupled 1D/2D models. (A) *boundary-connected*, (B) *internally-connected*, and (C) *vertically-connected* types. Blue areas indicate 2D domains, black dashed lines indicate 1D domains, and red crosses/arrows/lines indicate links between the models.

process with the identified internal constraints is then described in Section 4. Finally, application of the methodologies devel-
oped are demonstrated in Section 5, and the paper is concluded by providing a brief summary of the work and some possible
future directions are identified.

## 2  Overview of coupled 1D/2D hydrodynamic domains

Consider a hydrodynamic model domain $\Omega_{\mathrm{2D}} \subset \mathbb{R}^2$ defined by a simple (i.e., not self-intersecting) polygon, possibly with holes.
A so-called mesh-size function $h : \Omega_{\mathrm{2D}} \to \mathbb{R}$ that assigns a "target" element size $\Delta = h(x,y)$ to each point $(x,y) \in \Omega_{\mathrm{2D}}$ plays
a fundamental role in the construction of a triangulation $\mathcal{T}_h$ of $\Omega_{\mathrm{2D}}$. Here, the mesh-size function $h$ is represented as a bilinear
interpolant constructed on a rectilinear "background" grid that consists of a set of points X defined such that $\Omega_{2D} \subset \mathrm{Conv}(X)$,
where $\mathrm{Conv}(X)$ denotes the convex hull of X. In the existing ADMESH+ framework, several factors can be considered in the
construction of the mesh-size function, including user-specified minimum and maximum element sizes, boundary curvature,
etc; see Conroy et al. (2012) for a complete description.
In our previous work, the polygonal domain $\Omega_{\mathrm{2D}}$ and the mesh-size function $h$ (together with its corresponding background
grid) have served as the two primary inputs to the mesh generation process of ADMESH+. In this work, the incorporation and
construction of a third primary input is described that is fundamental to the generation of suitable meshes for coupled 1D/2D
hydrodynamic models — namely, a set of internal constraints that consists of a set of line segments interior to $\Omega_{\mathrm{2D}}$ along
which edges in the 2D triangulation must be constrained. Three types of internal constraints are considered. The first type are
line segments that represent centerlines of "narrow" channels, i.e., those channels that cannot be accurately resolved under the
user-specified minimum element size. These line segments make up the aforementioned 1D hydrodynamic domain. The second
type are line segments that align with sub-grid scale topographic features/structures, such as "narrow" barrier islands, levees





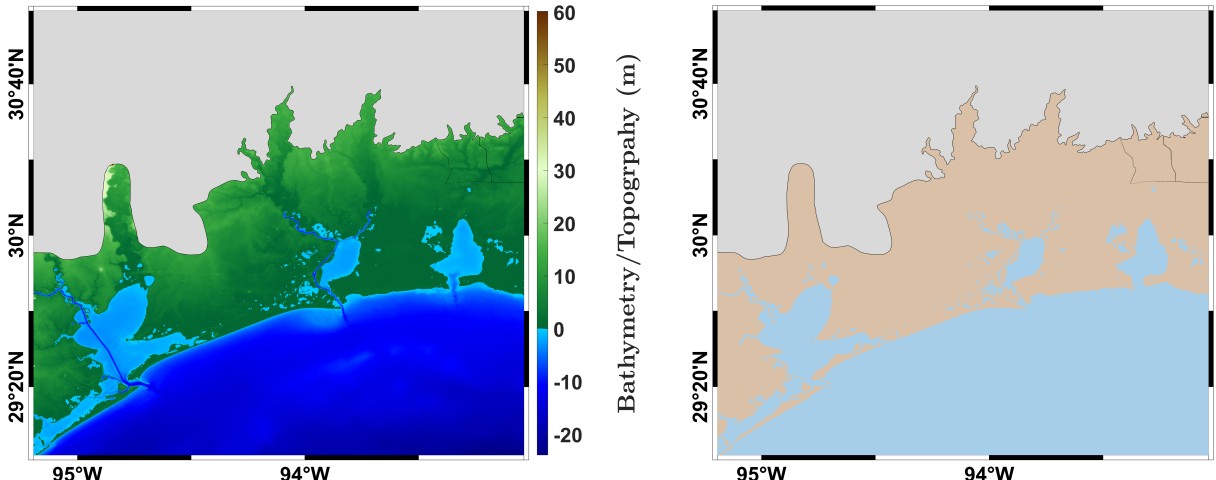

**Figure 2.** Example of a DEM defining the bathymetry/topography elevations (*left*) and the corresponding land-water mask (*right*), where $\Omega_{\mathrm{w}}$ is indicated by light blue.

and weirs, along which certain internal boundary conditions are enforced in the 2D hydrodynamic model; see, for example, Dawson et al. (2011). Finally, the third type of internal constraints are line segments that represent the boundary between the
land and water subdomains, i.e., the shorelines. Note that this type of internal constraint is neither part of the 1D domain nor an internal boundary but is desired in order to provide a clear distinction between the land and water subdomains by the generated 2D mesh.

We note that the described internal constraints can be provided directly from relevant data sources — for example, channel centerlines are available from the U.S. Geological Survey (USGS) National Hydrography Dataset (NHD) (U.S. Geological
Survey (2016)) — but are more generally obtained based on two functions defined over $\Omega_{\mathrm{2D}}$. One is a function $f : \Omega_{\mathrm{2D}} \to \mathbb{R}$ that assigns the (bare) earth surface elevation $z = f(x, y)$ to a point $(x, y) \in \Omega_{\mathrm{2D}}$. This is typically provided by a combination of topographic and bathymetric (gridded) DEMs. A second is a so-called indicator function $\mathbb{1}_{\Omega_{\mathrm{w}}} : \Omega_{\mathrm{2D}} \to \{0, 1\}$ of a subset $\Omega_{\mathrm{w}} \subset \Omega_{\mathrm{2D}}$ defined as

$$\mathbb{1}_{\Omega_{\mathrm{w}}}(x, y) = \begin{cases} 1 & \text{if } (x, y) \in \Omega_{\mathrm{w}} \\ 0 & \text{if } (x, y) \notin \Omega_{\mathrm{w}} \end{cases},$$

which indicates whether a point $(x, y) \in \Omega_{\mathrm{2D}}$ is (initially) "wet" (1) or "dry" (0), i.e., a so-called land water mask. We refer to $\Omega_{\mathrm{w}}$ as the water subdomain of $\Omega_{\mathrm{2D}}$; see Fig. 2. This is typically provided by a data set of closed polygons whose interiors indicate the water subdomain; see, for example, Wessel and Smith (1996). Below, we describe our methodology for identifying internal constraints from the data sets that inform these two functions.





## 3 Identification of internal constraints

In this section, we present our methodology for extracting the three types of internal constraints described above. The methodologies vary for the land and water portions of the domain. The key of the extraction of open-channels from the land portion of the domain is drainage area computed using the gradient of the input DEM(s). In the water portion of the domain, central to the extraction of open-channels is the width measurement of relevant features of the domain. Note that this methodology is further applied to identify internal boundaries (the second type of internal constraints) from the land portion of the domain. The third

type of internal constraint—namely, the boundaries between the land and water domains — is obtained as a by-product from this methodology. As two distinct methodologies are applied, they are described in separate subsections below.

### 3.1 Open-channels from the land subdomain

In the land subdomain, only the first type of internal constraint mentioned above, namely, channel centerlines, are identified. From input DEM(s), internal constraints representing "dry" channels (flow paths) in the land subdomain are detected by

integrating the esiting ADMESH+ code with TopoToolbox — a widely used, Matlab-based topographic toolbox (Schwanghart and Kuhn (2010)).

The channel detection algorithm in TopoToolbox is a flow accumulation algorithm based on the gradient of the DEM. At each grid point of the DEM, flow direction is computed for 8-connected neighbors to create a global flow direction matrix $\mathbf{M}$. Starting with a uniform unit water depth on each grid point of the DEM, expressed as a vector $\mathbf{w}$, the global flow direction

matrix is multiplied with the water depth vector to compute water depth at the next time step

$$\mathbf{w}^{i+1} = \mathbf{M}\mathbf{w}^i, \tag{1}$$

where $\mathbf{w}^i$ is water depth vector at $i$-th iteration. This operation is repeated until water completely leaves the system. Then, drainage area at each grid point is defined as the summation of water depth over all iterations:

$$\mathbf{a} = \mathbf{w}^0 + \mathbf{w}^1 + \cdots + \mathbf{w}^n, \tag{2}$$

where $\mathbf{a}$ is drainage area vector. Finally, the channels are identified by a threshold of minimum drainage area. Note that while the algorithm is developed based on an iteration method, in TopoToolbox, the drainage area is computed directly by using geometric procession:

$$\mathbf{a} = \mathbf{w}^0 + \mathbf{w}^1 + \cdots + \mathbf{w}^n + \cdots \tag{3}$$

$$= (\mathbf{I} + \mathbf{M} + \mathbf{M}^2 + \cdots + \mathbf{M}^n + \cdots)\mathbf{w}^0 \tag{4}$$

$$= (\mathbf{I} - \mathbf{M})^{-1}\mathbf{w}^0. \tag{5}$$

As mentioned above, while other datasets of open-channel centerlines (for example, USGS NHD) can be used, in our experience, open-channels extracted from TopoToolbox show better alignment with input DEMs.





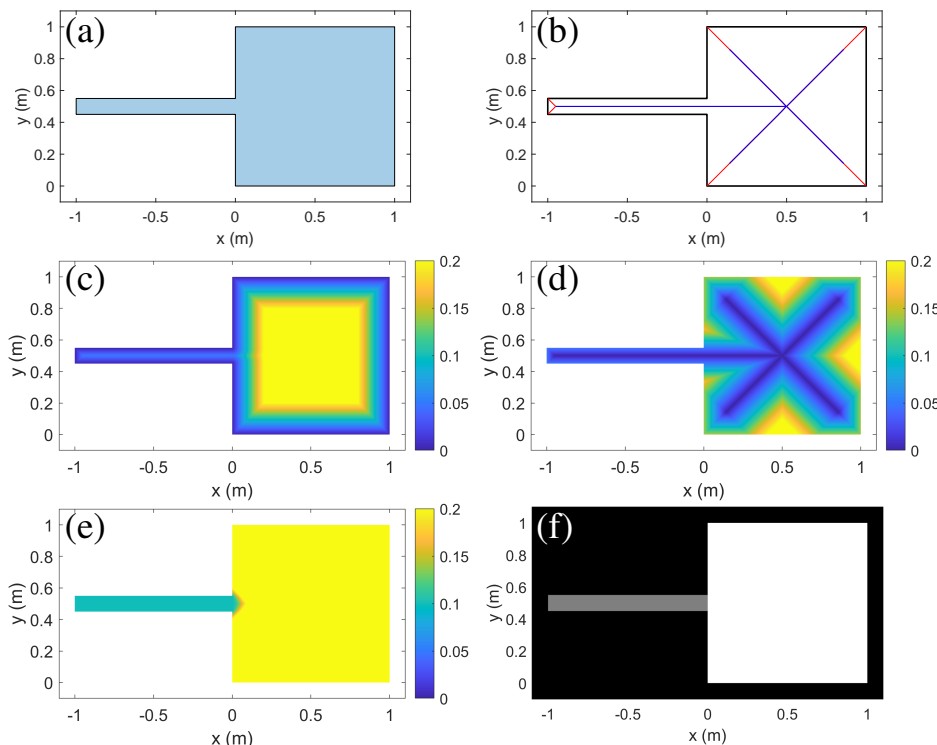

**Figure 3.** Process of mask decomposition into "narrow" and "wide" regions: (a) A given water mask. (b) The medial axis before/after pruning (*red/blue*). (c) The distance function to the boundary. (d) the distance function to the medial axis. (e) The width function. (f) The mask decomposition with level 1 (*gray*, "narrow" regions) and level 2 (*white*, "wide" regions).

## 3.2 Open-channels from the water subdomain and internal boundaries from the land subdomain

The water subdomain exhibits a wide range of scales from "large" open-water bodies to "narrow" channels and islands, the latter of which are represented as holes in the water mask. Towards the purpose of identifying narrow channels and small islands in the water subdomain, we describe our methodolgy for performing a width-based decomposition of the water subdomain that make use of a user-defined minimum width $\delta_w$, i.e., features of width $< \delta_w$ are identified as "narrow."

### 3.2.1 Width-based mask decomposition

We describe the step-by-step procedure for determining the width-based mask decomposition with the aid of the simple example shown in Fig. 3. For masks with more complex boundaries and/or holes, additional processes are required, which are described in Section 3.2.2.

**Step 1:** *For given polygon (Fig. 3 (a)), find the medial axis (Fig. 3 (b)).*



The medial axis $MA(P)$, of any closed polygon $P$, is defined as the set of interior points that have equal distances to two or more points on the boundary of $P$. Our computation of the medial axis is based on the vector distance transform (VDT) originally proposed by Mullikin (1992). Given a polygon $P$ with boundary $\partial P$, the value of the VDT, $V(\mathbf{x}, P)$, at a point $\mathbf{x} = (x, y) \in P$ is defined as

$$\mathbf{V}(\mathbf{x}, P) = \Phi(\mathbf{x}, \partial P) - \mathbf{x} \tag{6}$$

where

$$\Phi(\mathbf{x}, \partial P) = \underset{\mathbf{y} \in \partial P}{\arg\inf} \, \|\mathbf{x} - \mathbf{y}\|. \tag{7}$$

Using the VDT, the medial axis of $P$ can be obtained as (see Appendix A)

$$MA(P) = \{\mathbf{x} \in P : \nabla \cdot \mathbf{V}(\mathbf{x}, P) > 0\}. \tag{8}$$

Note that the VDT defined by (6) is not well-defined on the medial axis points as multiple arguments of the infimum are produced from (7). In numerical implementation, we arbitrarily choose one of the arguments of infimum of (7).

**Step 2.** *Prune the medial axis (Fig. 3 (b)).*

After obtaining the medial axis, pruning near "sharp" corners is required to improve the quality of the width function, which is described later. First, we construct a hierarchy of MA branches, i.e., branches with free-ends are order 1, branches connected to order 1 are order 2, etc. Next, for each MA point $\mathbf{p}$ on order 1 branches, we define width and angle of MA as

$$\ell = \max_{i=1,\cdots,4} \|(\mathbf{p} + \mathbf{v}) - (\mathbf{p}_i + \mathbf{v}_i)\| \tag{9}$$

and

$$\theta = \max_{i=1,\cdots,4} \cos^{-1}\left(\frac{\mathbf{v} \cdot \mathbf{v}_i}{\|\mathbf{v}\| \|\mathbf{v}_i\|}\right), \tag{10}$$

where $\mathbf{p}_i \, (i = 1, \cdots, 4)$ are four neighboring background grid points of $\mathbf{p}$, and $\mathbf{v}$ and $\mathbf{v}_i$ are VDTs corresponding to $\mathbf{p}$ and $\mathbf{p}_i$, i.e., $\mathbf{v} = \mathbf{V}(\mathbf{p}, P)$ and $\mathbf{v}_i = \mathbf{V}(\mathbf{p}_i, P)$ (see Fig. 4). As mentioned in the previous step, the VDTs on the MA are arbitrarily chosen in numerical implementation. This may cause differences for lengths of the MA, but these differences are insignificant (see Fig. 4). Note that inverse cosine function computed with `acos` function in MATLAB gives $\theta_i \in [0, \pi]$. Finally, the MA points near corners are identified and "pruned" based on the thresholds

$$\theta < \delta_\theta \quad \text{and} \quad \ell < \delta_\ell, \tag{11}$$

where $\delta_\theta$ and $\delta_\ell$ are specified threshold values used for pruning. Based on our experiments, the values $\delta_\theta = 0.9\pi$ and $\delta_\ell = 2\delta_w$ provide reasonable results.





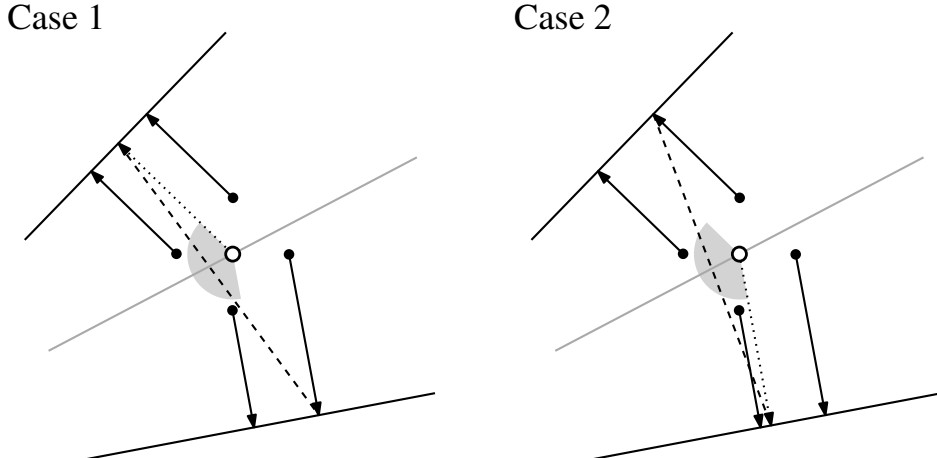

**Figure 4.** Schematic of computation of width and angle (*dashed lines* and *gray circular sectors*) of medial axis (*gray solid line*). *White dots* are given medial axis point, *black dots* are neighboring background grid points, and *solid arrows* are VDTs from neighboring background grid points. Two possible choices of VDT are marked with *dotted arrows* in Case 1 and 2.

**Step 3.** *Compute distance functions and define the width function (Fig. 3 (c), (d), and (e)).*


Now we define two distance functions, one measures the closest distance from a point $\mathbf{x}$ to the boundary $d(\mathbf{x}, \partial P)$ and the other measures the closest distance from a point to the $\mathbf{x}$ medial axis $d(\mathbf{x}, MA(P))$. The former can be measured using the VDT:

$$d(\mathbf{x}, \partial P) = \|\mathbf{V}(\mathbf{x}, P)\|. \tag{12}$$

Then, the width function is defined as twice the sum of the two distance functions (see Fig. 5), i.e.,

$\quad f_w(\mathbf{x}, P) \coloneqq 2(d(\mathbf{x}, \partial P) + d(\mathbf{x}, MA(P))). \tag{13}$

Note that, without pruning the medial axis, the width function results in $f_w(\mathbf{x}) \approx 0$ in the vicinity of every corner. This is because both boundary lines and the (unpruned) medial axis exist at every corner of the polygon boundary, and thus $d(\mathbf{x}, \partial P) \approx 0$ and $d(\mathbf{x}, MA(P)) \approx 0$. This results in the area around every corner being identified as "narrow," even though they are corners of large water bodies. This is the rational for pruning the medial axis around corners, which was presented in Step 2.


**Step 4.** *Decompose the polygon with a user-defined minimum width $\delta_w$ (Fig. 3 (f)).*

With the width function Eq. 13, the given polygon $P$ can be decomposed as:

$$P_1 \coloneqq \{\mathbf{x} \in P : f_w(\mathbf{x}, P) < \delta_w\}, \tag{14}$$

and

$$P_2 \coloneqq \{\mathbf{x} \in P : f_w(\mathbf{x}, P) \geq \delta_w\}. \tag{15}$$





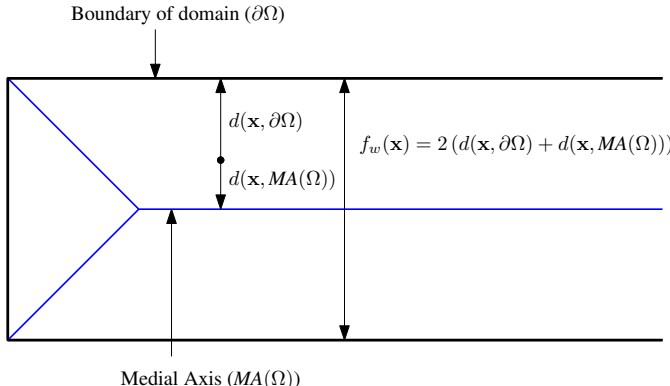

**Figure 5.** Schematic of width function.

We call the masks $P_1$ and $P_2$ level 1 and 2 masks, respectively. Note that the level 1 and 2 masks can be used for 1D and 2D domains for simple cases, but additional processes are required for complex geometries. The level 1 and 2 masks are not identical with 1D and 2D domain in that case, and thus we use the term "level" to avoid confusion.

### 3.2.2 Additional processes for complex geometries

The challenge in real applications comes from complex boundaries and the presence of small "narrow" islands within large water bodies. The complexities result in "noise" of the width function and do not provide a clear distinction between level 1 and 2 masks, and additional processes are applied to resolve the noise. In order to catch the narrow islands, the mask decomposition needs to be applied to the land mask in a similar way as it is to the water mask decomposition. Applying the mask decomposi-

tion for both land and water masks also requires an additional process, which is described in this section. Hereinafter, the land and water masks are denoted by $M_L$ and $M_W$, respectively. These processes are described below in the context of an example (see Fig. 6 and 7).

**Step 1.** *Remove "small" islands (Fig. 6 (a) and (b)).*


Small islands, by which we mean islands with areas that are much smaller than the square of minimum element size, are removed at the first step for two reasons. First, it is expected that small islands will not have significant effects on the hydrodynamic models. Second, mask decomposition with small islands tends to result in "poor" internal constraints. Mask decomposition converts small islands to internal constraints with length of their major axis, and therefore small islands tend to create

internal constraints shorter than the minimum element size. Note that short internal constraints can be identified and removed after decomposing masks as well. In this example, islands with area less than 1,000 m$^2$ are removed, where the minimum element size is set as 45 m (see Fig. 6 (a) and (b)).



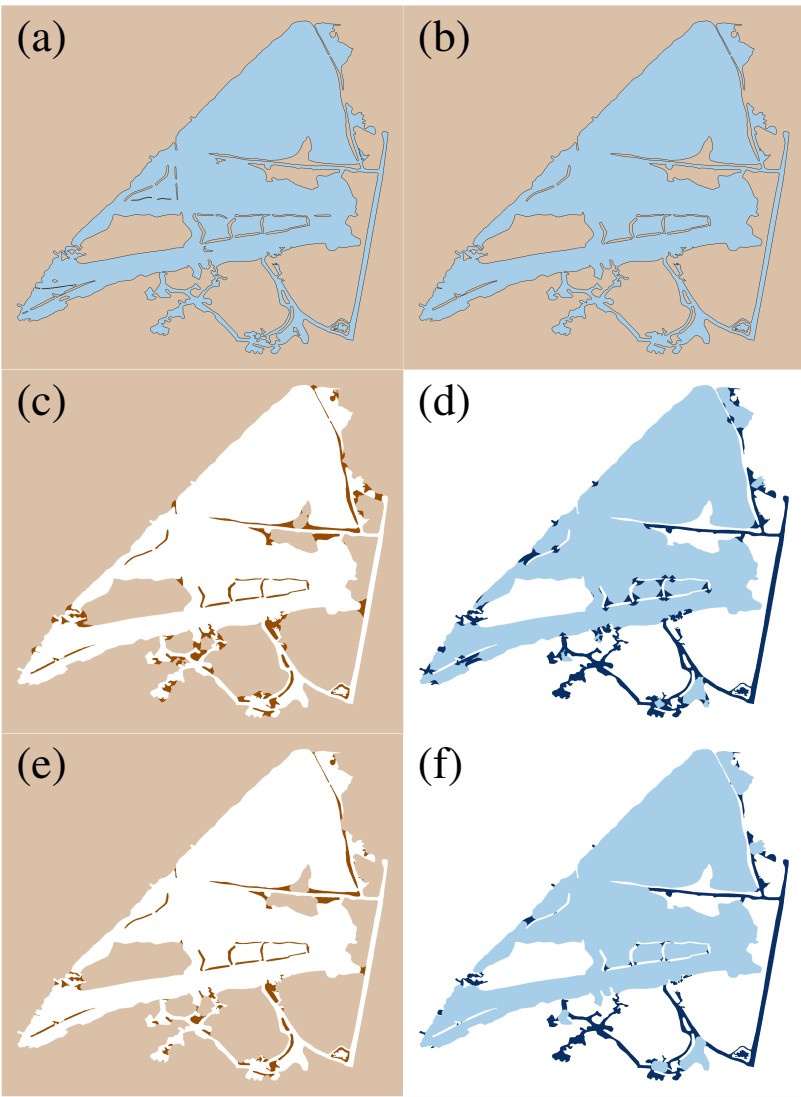

**Figure 6.** Example of the step-by-step procedure for complex geometries applied to part of the Lower Neches Basin, TX: (a) Original land/water masks from the input data set (*brown/blue* areas, respectively). (b) Land/water masks after removing "small" islands (Step 1). (c,d) Decomposed land/water masks consisting of level 1 and 2 land masks (*dark brown* and *light brown* areas, respectively) and level 1 and 2 water masks (*dark blue* and *light blue* areas, respectively). (e,f) Updated land/water masks after filling (Step 2).



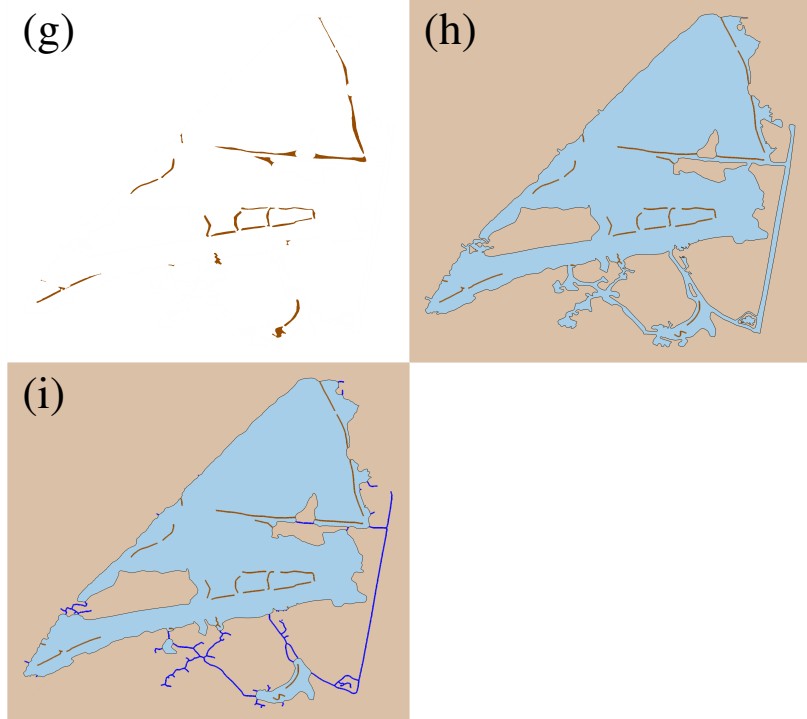

**Figure 7.** Example of the step-by-step procedure for complex geometries applied to part of the Lower Neches Basin, TX (*continued*): (g) Regions of level 1 land mask surrounded by level 2 water mask (Step 3). (h) Updated land/water masks after transferring regions in (g) to internal boundary constraints (*brown lines*) (Step 4). (i) Final land/water masks including open-channel constraints (*blue lines*) after applying mask decomposition for new water mask (Step 5).

**Step 2.** *Apply width-based decomposition with filling to land and water masks (Fig. 6 (e) and (f)).*


While the mask decomposition gives a clear distinction between "narrow" and "wide" regions in the simple example (see Fig. 3), more complex cases can result in noise in the width function, which can be removed through a so-called *filling* method.

The filling method is based on the idea of the maximal disk (see Fig. 8 (a)), which is defined as

$$D(\mathbf{x}, P) = \{\mathbf{y} \in P : \|\mathbf{x} - \mathbf{y}\| \le d(\mathbf{x}, \partial P)\}. \tag{16}$$

And the level 1 and 2 masks are updated with the maximal disks centered in level 2 masks

$$\tilde{M}_{i2} := M_{i2} \cup \mathbf{D}_i \quad \text{and} \quad \tilde{M}_{i1} := M_{i1} \setminus \mathbf{D}_i, \tag{17}$$

where

$$\mathbf{D}_i = \{D(\mathbf{x}_j, M_i) \text{ for all } \mathbf{x}_j \in MA(M_i) \cap M_{i2}\} \tag{18}$$





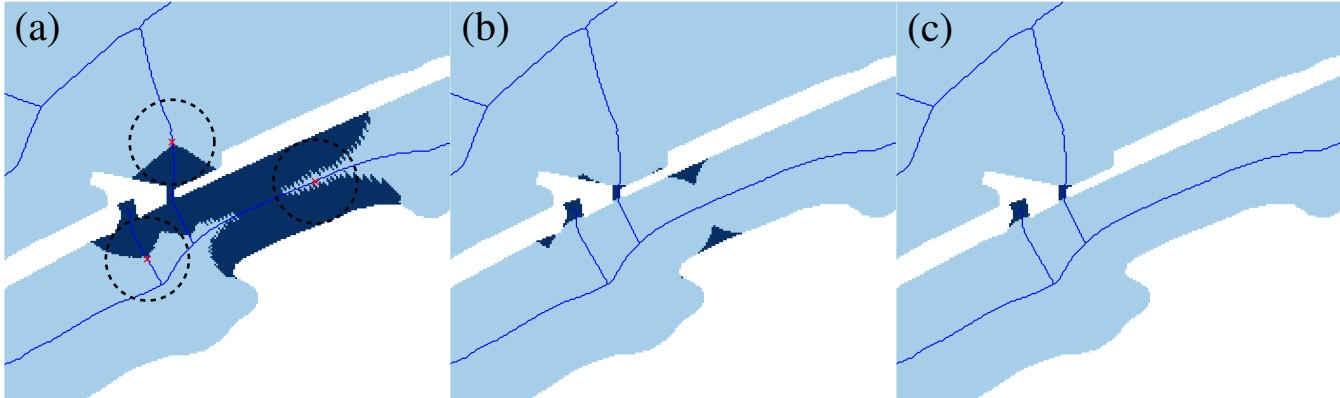

**Figure 8.** Example of the step-by-step process of filling level 2 mask: (a) Maximal disks (*black dashed lines*) whose centers (*red cross marks*) are in level 2 water masks. (b) Level 1 and 2 water masks after filling. (c) Level 1 and 2 water masks after transferring level 1 regions without any MA to level 2 masks. (*Blue lines* are the (pruned) medial axes, *light blue areas* are level 2 water masks, and *dark blue areas* are level 1 water masks.)

and $M_i = M_L$ and $M_W$. Note that the maximal disks are sought on level 2 masks only. Also, there are some regions of $\tilde{M}_{i1}$ that
do not include any MA (see Fig. 8 (b)). These regions are redundant as they cannot be represented by the MA. Therefore, we transfer such regions to level 2 masks and update the masks (see Fig. 8 (c)).

$$M_{i2} \leftarrow \tilde{M}_{i1} \cup \mathbf{R}_i \tag{19}$$

$$M_{i1} \leftarrow \tilde{M}_{i1} \setminus \mathbf{R}_i \tag{20}$$

where

$$\mathbf{R}_i = \{P \in \tilde{M}_{i1} : P \cap MA(M_i) = \emptyset\} \tag{21}$$

Note that with these procedures the level 2 masks have a smoother boundary and provide a better representation of the large water bodies; see panels (e) and (f) of figure 6, which are updated masks of panels (c) and (d), respectively, after filling.

**Step 3.** *Find regions of level 1 land mask that are surrounded by level 2 water mask (Fig. 7 (g)).*


The purpose of this step is to identify regions of level 1 land mask that are retained as internal constraints in the 2D hydrodynamic model. These "narrow" land regions are typically modeled as internal boundaries over which simple sub-grid scale flow parameterizations are performed, using, for example, simple weir-based formula, see, for example, Dawson et al. (2011).

First, the thinness of land regions is determined by the so-called isoperimetric ratio (IPR), which is defined as $\text{Perimter}^2/\text{Area}$.
Note that the IPR is a dimensionless number, which is higher for thinner regions. Here, we set a threshold of 30. Second, in order to identify if each region is surrounded by level 2 water mask, we first set a buffer for each region. The buffer size is set





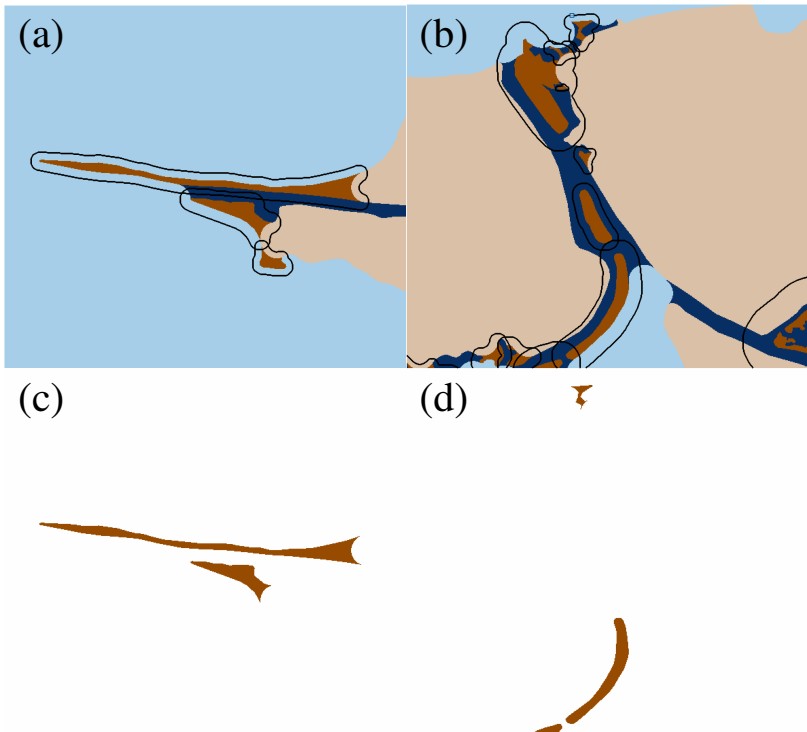

**Figure 9.** Example of the selection of level 1 land regions that will serve as internal constraints: (a,b) Example of buffers (*black lines*) of level 1 land regions (*dark brown* areas) and (c,d) Selected level 1 land regions. *Light brown* areas are level 2 land masks, and *dark blue* and *light blue* areas are level 1 and 2 water masks, respectively.

as half of the length of the minor axis of the ellipse that has the same normalized second central moments as the region (see Fig. 9 (a) and (b)). The minor axis length is computed using `regionprops` function in MATLAB. We then check the area of the level 2 masks within the buffer. We define the land region to be surrounded by level 2 water mask if the area of level 2 water mask is greater than twice the area of level 1 land mask within the buffer (see Figure 9 (c) and (d), which show selected regions of level 1 land mask shown in Figure 9 (a) and (b), respectively). Note that level 1 water masks will be internal constraints (channel centerlines). This means that if a region of level 1 land mask is surrounded by level 1 water mask, then there are too many internal constraints too close each other. Therefore, we only retain regions of level 1 land mask that are surrounded by level 2 water mask, denoted by $M_{\mathrm{L1W2}}$, which can be expressed as

$$M_{\mathrm{L1W2}} = \{P \in M_{\mathrm{L1}} : \mathrm{IPR}(P) > 30 \text{ and } \frac{||M_{\mathrm{W2}} \cap B(P)||}{||M_{\mathrm{L2}} \cap B(P)||} > 2\}, \tag{22}$$

where $\mathrm{IPR}(P)$ is the isoperimetric ratio, $B(P)$ is the buffer of region $P \in M_{\mathrm{L1}}$, and $||\cdot||$ denotes area.

**Step 4.** *Transfer regions of level 1 land mask identified in Step 3 to the water mask (Fig. 7 (h)).*





As described in Step 3, the narrow regions identified will now serve as internal constraints (specifically, internal boundaries as described above) in the mesh generation process and no longer need to be in land mask. Therefore, these regions are transferred into the water mask and "updated" land and water masks are defined as

$$M_{\mathrm{L}}^* := M_{\mathrm{L}} \setminus M_{\mathrm{L1W2}} \tag{23}$$

$$M_{\mathrm{W}}^* := M_{\mathrm{W}} \cup M_{\mathrm{L1W2}}. \tag{24}$$

Note that the centerlines of narrow land regions are used as internal constraints (internal boundaries) for mesh generation:

$$\mathbf{s}_1 = MA(M_{\mathrm{L}}) \cap M_{\mathrm{L1W2}}. \tag{25}$$

**Step 5.** *Apply the width-based mask decomposition to the new water mask (Fig. 7 (i)).*

The width-based decomposition of the water mask that now includes the narrow land regions described above will be different from the width-based decomposition of the original input water mask. Therefore, the width-based decomposition must be applied again. Note that, in this step, the decomposition for the land mask is not required. The centerlines of updated level 1 water mask $M_{\mathrm{W1}}^*$ form the domain of 1D hydrodynamic model, i.e.,

$$\Omega_{\mathrm{1D}} := MA(M_{\mathrm{W}}^*) \cap M_{\mathrm{W1}}^*. \tag{26}$$

Here, we have the second type of internal constraint (1D domain) for mesh generation

$$\mathbf{s}_2 = \Omega_{\mathrm{1D}}. \tag{27}$$

The domain of the 2D model is the entire domain except the domain of 1D model. Note that it includes the updated level 1 mask as well as the updated level 2 water mask and land mask, i.e.,

$$\Omega_{\mathrm{2D}} := M_{\mathrm{L}}^* \cup M_{\mathrm{W2}}^* \cup (M_{\mathrm{W1}}^* \setminus \Omega_{\mathrm{1D}}). \tag{28}$$

It is preferred that the mesh elements are aligned along boundaries between water bodies and land. This can be ensured by passing the boundary of $M_{\mathrm{W2}}^*$ as internal constraints even though it is neither an open-channel or an internal boundary, which is the third type of internal constraint for mesh generation:

$$\mathbf{s}_3 = \partial M_{\mathrm{W2}}^*. \tag{29}$$

**Step 6.** *Construct mainstreams of 1D domain and internal boundaries.*



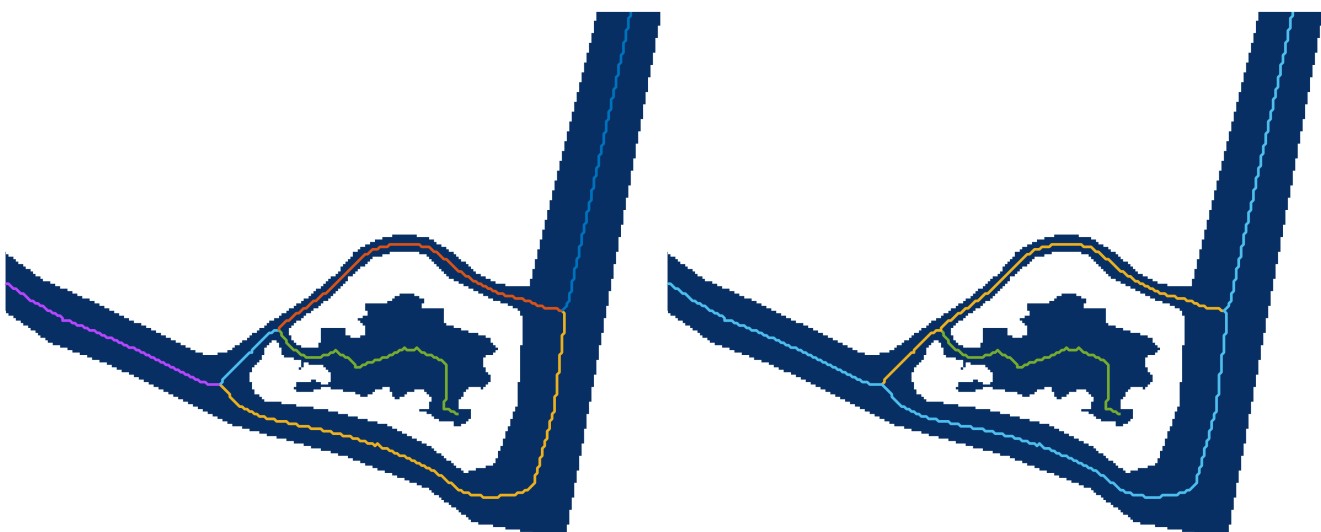

**Figure 10.** Example of construction of mainstream for 1D domain. Channel network before and after mainstream construction (*left* and *right*, respectively). The colors of lines indicate different segments.

The 1D domains and internal boundaries contain the centerlines of narrow regions of the water/land mask. These centerlines are obtained from the medial axis of the mask, which has been pruned and ordered into hierarchic medial axis branches. This branch-wise ordering creates several "short" MA segments (see, for example, Figure 10), which is undesirable for computing the internal constraints curvature that is used to help determine the size of the 1D elements, as described in the next section. Therefore, a procedure to construct channel "mainstreams" is applied as follows. First, at each joint, the pair of segments, or branches, that have minimum (absolute) curvature are merged together to form a new segment (the mainstream). If there are more than three branches at a joint, another pair of branches with minimum curvature, excluding the mainstream, are merged and set as a sub-stream. The sub-streams are collected until there is no pair at a joint.

To conclude this section, the algorithm described generally provides good identification of "narrow" regions and the three types of internal constraints. However, this identification is not perfect and can result in some "narrow" regions being falsely identified, or misclassified, as level 2 regions. There are two possible reasons for this misclassification. The first one is related to the quality of the medial axis calculation. Recall that the medial axis is obtained by computing the divergence of the VDT and is subsequently pruned based on specified tolerances. This approach can result in an inaccurate estimation of the width function, especially within small regions. The second reason relates to the use of the dimensionless width parameter. Specifically, the identification of level 1 land regions surrounded by level 2 water masks (as described in Step 3) depends on the IPR threshold and the ratio of the surrounding level 2 water/land mask. This can result in some level 1 land regions being omitted, which are desired to be represented by their centerlines.





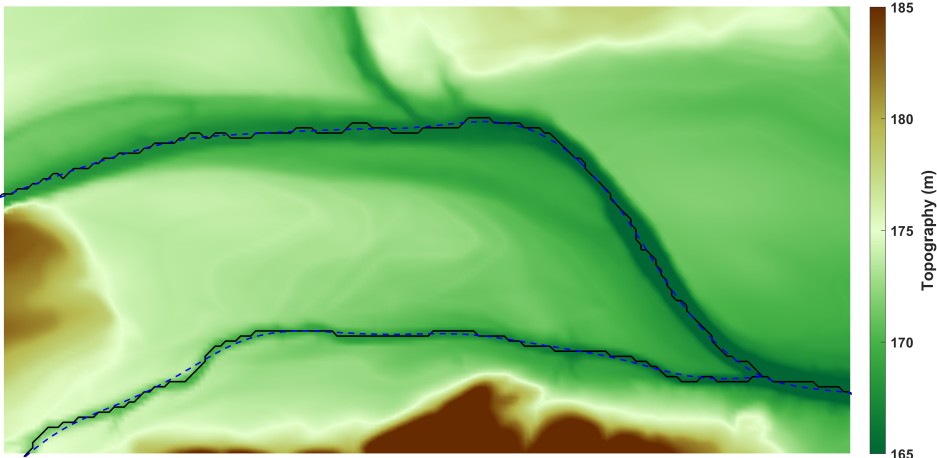

**Figure 11.** Example of extracted channel centerlines with TopoToolbox (*black solid line*) and smoothed line (*dashed blue line*, with RMSE$_{\text{desired}}$ = 10 m).

Additionally, there can be some internal constraints, even if they are correctly identified, that result in elements that are too small or of poor quality. For example, there can be internal constraints that are too close to each other. Note that, by choosing level 1 land regions surrounded by level 2 water mask (in Step 3), it is unlikely that internal boundaries are too close to open-channels. However, there are some internal boundaries/open-channels which are too close to the third internal constraint, namely, boundaries between water bodies and land.

While these problems are related to identification of "narrow" and "wide" regions and internal constraints, it is easier to resolve them during the mesh generation process itself. Therefore, treatment of the problems mentioned here will be described in the next section (specifically, see Section 4.4).

## 4  Force equilibrium with internal constraints

In this section, we introduce a methodology to generate 2D finite element meshes given the identified internal constraints. The goal of mesh generation with internal constraints is to use efficient mesh resolution along the internal constraints so that it preserves the geography of the study areas with reduced computational demand. This is achieved by assigning mesh size inversely proportional to the curvature of internal constraints. Also, additional processes are applied to ensure robustness of the force-equilibrium algorithm with internal constraints.

### 4.1  Smoothing internal constraints

The internal constraints, which are given by user-input or extracted from a DEM, are often based on a structured grid. This can produce, for example, a "choppy" set of channel centerlines (see Fig. 11) that do not provide a good basis for computing channel curvature, which is used in the process of determining mesh node placement and element size. Therefore, in order to provide



a smooth curve from which we can compute curvature, a cubic spline smoothing is applied based on the `csaps` function in MATLAB. The `csaps` function returns a smooth spline interpolation $f_p$ to the $N$ data points $(r_i, y(r_i))$, $i = 1, \cdots, N$ that minimizes

$$p \sum_{i=1}^{N} w_i |y_i - f_p(r_i)|^2 + (1-p) \int \lambda(t) |D^2 f_p(t)|^2 dt \tag{30}$$

where $p$ is a smoothing parameter, $D^2 f_p$ is the second derivative of $f_p$, and $w_i$ and $\lambda$ denote error measure weights and a weight function, respectively (see the MATLAB `csaps` help documentation for more details). Note that internal constraints are two-dimensional curves $\mathbf{s} = (x_i, y_i)$. Thus, for each internal constraint, we define a parametric curve as $(x(r_i), y(r_i))$, where $r_i = i$ for $i = 1, \cdots, N$, and find smoothed curves $\tilde{x}_p$ and $\tilde{y}_p$ individually. Note that the parameter $r_i$ is a set of arbitrary values, and the `csaps` function depends not only on the smoothing parameter $p$ but also on the parameter $r_i$. In order to get standardized smoothing, we find a smoothing parameter with root mean square error (RMSE) closest to user-defined RMSE:

$$p^* = \operatorname*{argmin}_{p} \left| \sqrt{\sum_{i=1}^{N} \frac{(x(r_i) - \tilde{x}_p(r_i))^2 + (y(r_i) - \tilde{y}_p(r_i))^2}{N}} - \text{RMSE}_{\text{desired}} \right|. \tag{31}$$

Then a smoothed curve is defined as

$$\tilde{\mathbf{s}} = (\tilde{x}_{p^*}, \tilde{y}_{p^*}). \tag{32}$$

Note that the user-defined RMSE ($\text{RMSE}_{\text{desired}}$) should be carefully selected. If it is too high, the smoothed curve is close to a straight line. If it is too low, it does not give enough smoothing. In our numerical experiments, a smoothing with $\text{RMSE}_{\text{desired}}$ between 1 and 10 meters is generally appropriate in representing the "overall" curvature of the 1D constraints (see Fig. 11).

## 4.2 Initial target mesh size and 2D gradient limiting

Given a smoothed internal constraint segment $\tilde{\mathbf{s}}_i$, initial target mesh sizes along the curve are computed by

$$\tilde{h}_{2D}(\mathbf{s}_i) := \frac{1}{K |\kappa(\tilde{\mathbf{s}}_i(r))|}, \, i = 1, \cdots, N, \tag{33}$$

where $K$ is the number of elements per radian (a user-defined parameter) and $\kappa(\tilde{\mathbf{s}}_i(r))$ is the curvature of smoothed internal constraints $\tilde{\mathbf{s}}_i(r)$.

It is often desired to ensure that the 2D element sizes grade properly in the final mesh. Common approaches to achieve this are marching methods (e.g., Persson (2006); Roberts et al. (2019)) and gradient limiting (e.g., Conroy et al. (2012); Persson (2006)). In this paper, we adopt the gradient limiting approach. Briefly, we find a steady-state solution to the so-called gradient limiting equation:

$$\frac{\partial h}{\partial t} + |\nabla h| = \min(|\nabla h|, g), \tag{34}$$

where $g$ is related to a user-defined parameter that controls the ratio of neighboring element sizes (see Conroy et al. (2012) for details). The gradient limiting equation is solved with the initial condition

$$h(\mathbf{x}, t = 0) = h_0(\mathbf{x}). \tag{35}$$





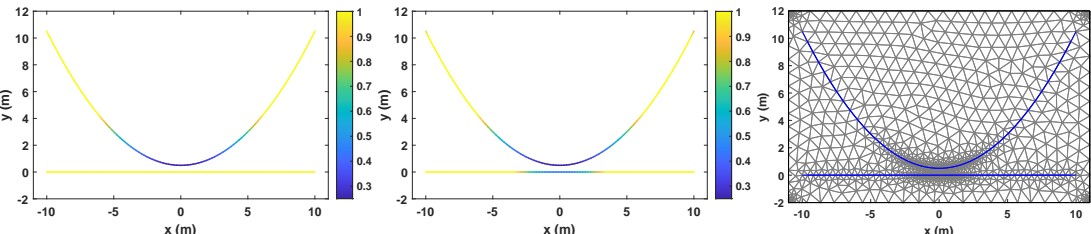

**Figure 12.** Example of 2D gradient limiting on internal constraints. Target mesh size without and with gradient limiting (*left* and *center*, respectively), and the 2D mesh generated with given internal constraint (*right*).

The gradient limiting is applied with the initial mesh size, $h_0(\mathbf{x})$, on internal constraints being defined by Eq. 33.. Note that a one-dimensional gradient limiting equation could be solved for each internal constraint. However, this may result in inappropriate target mesh sizes if internal constraints are close to each other (see Fig. 12).

### 4.3 Generation of 1D meshes on internal constraints

In order to generate collocated 1D elements and 2D edges, 1D meshes are first generated on internal constraints and then used as fixed points in the 2D mesh generation. The target element sizes of the 1D meshes on each internal constraint are defined by projecting the gradient limited mesh size $h_{2\mathrm{D}}$, which is the solution of Eq. 34 with $\tilde{h}_{2\mathrm{D}}$ as initial condition. i.e.,

$$h_{1\mathrm{D}}^i(r) = h_{2\mathrm{D}}(\mathbf{s}_i(r)), \ i = 1, \cdots, N. \tag{36}$$

Then, applying 1D force equilibrium with the target size on each internal constraints provides 1D nodes $r_i, i = 1, \cdots, N$. Now, fixed points of the 2D mesh generation are defined by

$$\mathbf{x}_i = \mathbf{s}(r_i) = (x(r_i), y(r_i)), \ i = 1, \cdots, N. \tag{37}$$

Note that the positioning of the fixed points is based on the original internal constraints $\mathbf{s}(r)$ instead of smoothed curves $\tilde{\mathbf{s}}(r)$, because smoothing can result in deviations from the original set of points defining the internal constraints as described in Section 4.1. Also, it is required to keep junctions of the curves so that the physical connections are not missed. This can simply be ensured by using junction points as fixed points of the 1D force equilibrium.

### 4.4 Post-processes for 1D mesh

As noted in Section 3.2.2, post-processes for the generated 1D meshes are applied to improve identification of internal constraint types and to improve 2D mesh quality.

Note that regions of the level 2 land mask $M_{\mathrm{L2}}$ and the updated level 2 water mask $M_{\mathrm{W2}}^*$ represent "wide" regions that will be represented with 2D elements. In order to represent such regions with 2D elements, there should be at least three 1D elements along the boundaries of the level 2 regions. However, for the falsely identified, as level 2, regions, their perimeter is not long enough to have three or more 1D elements. This is likely to happen for regions with small areas that are round in





shape. The IPR is lower for these types of areas, which results in those regions not being selected with the IPR filter when level 1 land regions surrounded by level 2 water mask are identified (see Eq. 22). Again, the boundaries of the level 2 regions are the third type of internal constraints. When 1D meshes are generated along the internal constraints corresponding to the falsely identified regions, there is only one element per region and such 1D elements are transferred to the first or second type
of internal constraints.

As also noted in Section 3.2.2, there can be some open-channels/internal boundaries, the first and second type of internal constraints that are too close to boundaries between water bodies and land, the third type of internal constraints. This will result in 2D elements that are too small between such internal constraints. Therefore, if the all 1D mesh nodes on an internal constraint are closer than $h_{\min}/2$ to any other 1D mesh on third type of internal constraint, then the 1D meshes are removed.
As we fixed junction points to preserve physical connections, there can be 1D mesh node clusters, which are sets of 1D mesh nodes located within $h_{\min}/4$ each other. Since this will result in 2D elements that are too small, the 1D mesh nodes are merged into the centroid of the cluster.

Finally, note that the mesh size of 1D elements (the distance between the 1D nodes in Eq. 37) is not identical to the target mesh size provided by Eq. 36. The mesh generated from the force equilibrium algorithm has mesh sizes "relative" to target mesh sizes; see Persson and Strang (2004) for details. Due to the nature of force equilibrium algorithm, there can be 1D
elements whose length is shorter than $h_{\min}/2$. In order to obtain high quality 2D elements, such 1D elements are removed.

### 4.5  Generation of 2D meshes with fixed points

The 2D force equilibrium is applied after 1D meshs are generated, by adopting 1D mesh nodes as *fixed points*. Again, note that the mesh size of the 1D mesh (the distance between the 1D nodes in Eq. 37) is not identical to the target mesh size Eq. 36. Due
to the discrepancy between the target mesh size (of the 2D mesh) and the actual mesh size (of the 1D mesh), there are some "non-converging" nodes near internal constraints. Note that, in general, the displacement of nodes decreases during the force equilibrium iterations and nodes converge to their final node locations. However, the non-converging nodes keep moving back and forth near internal constraints, and an additional treatment is applied to resolve the non-converging node situation.

This treatment consists of the following steps (see Fig. 13): 1) Find nodes near the internal constraints that are not fixed
points but have distances less than $h_{\min}/2$. 2) Compute the length of the closest 1D element to this node. 3) If the length of closest element is greater than $2h_{\min}$, then add the node to a new fixed point on the middle of the 1D element (Fig. 13 (a1) and (a2)). Otherwise, remove the node (Fig. 13 (b1) and (b2)).

This density control is applied while the 2D force equilibrium is being applied. However, note that there might be a number of nodes near internal constraints, which are located within $h_{min}/2$ at early stages of the force equilibrium process. Therefore,
the density control is applied for later stages of the force equilibrium, which starts from $0.8 \times$ maximum iteration number.





(a)

(b)

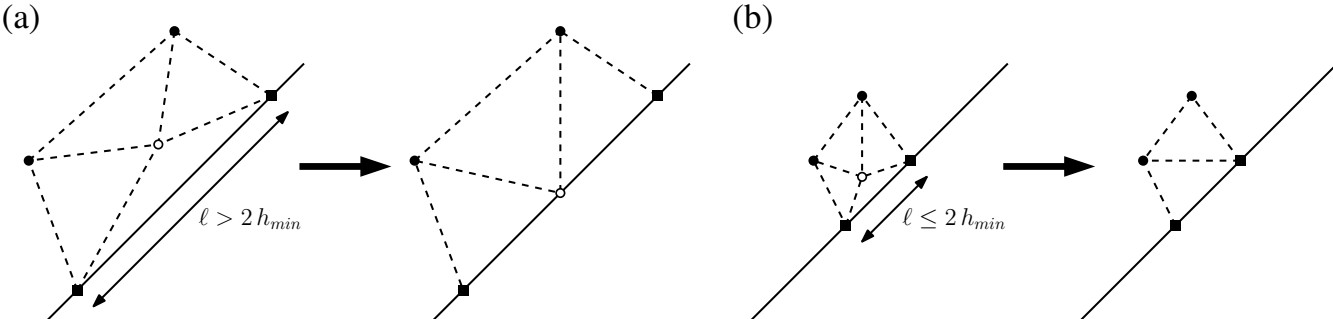

**Figure 13.** Schematic of density control. *Black squares* are 1D mesh nodes (fixed points in 2D force equilibrum algorithm), *black circles* are 2D mesh nodes, *white circles* are non-converging nodes, and *dashed lines* are triangulations. (a) If closest 1D mesh size is greater than $2h_{\min}$, non-converging node is added to 1D mesh node. (b) If closest 1D mesh size is shorter than $2h_{\min}$, the node is removed.

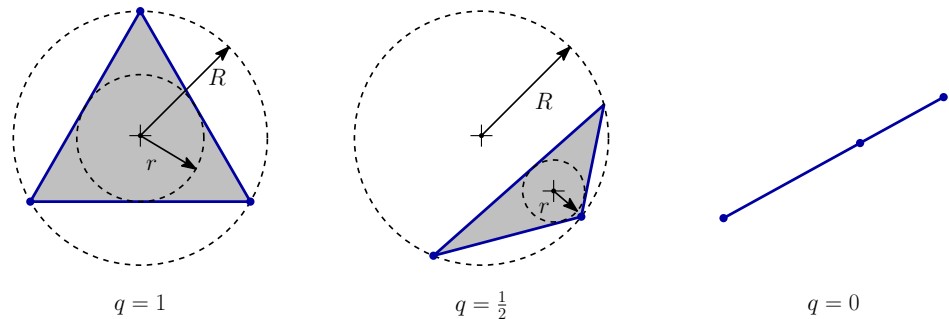

$q = 1$        $q = \frac{1}{2}$        $q = 0$

**Figure 14.** Geometric depiction of the element quality measure used to assess mesh quality.

## 5  Results

The mesh generation algorithm presented in this paper is applied to three test cases. The first two test cases are for inland watersheds without water subdomains. The third test case is applied for a coastal basin to highlight the performance of identification of 1D domains in water subdomains.

There are several measures used to assess the quality of a mesh (see Field (2000)). The measure used in this paper is twice the ratio of inradius $r$ and circumradius $R$ of each triangular element, i.e.,

$$q = 2\left(\frac{r}{R}\right) = \frac{(b+c-a)(c+a-b)(a+b-c)}{abc} \tag{38}$$

where $a, b$, and $c$ are the edge lengths of the triangular element. Note that this measure gives $q = 1$ for an equilateral triangle and $q = 0$ for a completely degenerate triangle (see Fig. 14)



## 5.1 The Middle Bosque River Watershed

The Middle Bosque River watershed (MBRW), located in central Texas, has been the subject of numerous computational hydrological studies; see, for example, Bailey et al. (2021); Park et al. (2019); Tefera and Ray (2023). Given this interest and the complex network of channels that must be represented for accurate model studies (see Figure 15), the MBRW presents an ideal test case for the developed 1D/2D mesh generation process described.

The MBRW covers an area of approximately 516 km$^2$ within the much larger Brazos River Basin ($\approx 119,174$ km$^2$) — the second largest river basin by area within Texas. The boundary of the MBRW is obtained from USGS Watershed Boundary Dataset (WBD) (U.S. Geological Survey (2014)), which provides the input (polygonal) domain $\Omega_{2D}$ for the mesh generation process. In addition to this input, a DEM covering the MBRW is available from the USGS 3D Elevation Program (3DEP) (U.S. Geological Survey (2017)), with the highest resolution available for the whole watershed being 1/3 arc second (approximately 10 meters). While channel centerlines are available for the MBRW from the aforementioned USGS National Hydrography Dataset (NHD), in this test case, the channel centerlines, which constitute the 1D domain $\Omega_{1D}$, are extracted within our mesh generator using TopoToolbox as describe in Section 3.1. The MBRW domain boundary, DEM, and extracted channels are shown in Figure 15.

Given the inputs described above, the mesh is automatically generated using the procedure outlined with the following user-defined parameters: Minimum and maximum elements sizes are set to 30 m and 500 m, respectively, the number of elements per radian $K$ in Eq. 33 is set to 20, the grading limit $g$ in Eq. 34 is set to 0.15, and the smoothing RMSE in Eq. 31 is set to 10 m. The resulting mesh is shown in Figures 16, 17, and 18, where several qualities of the mesh can be visually noted. First, the dashed blue lines of panels (a1) and (b1) of Figure 16 show close-ups of the smooth spline approximation of the channel centerlines that have been extracted from the input DEM. The accompanying panels (a2) and (b2) of Figure 16 show the node distribution of the 1D mesh that is generated along these channels, where it can be noted that smaller element sizes are present in highly curved areas and where elements sizes are relaxed in straighter channel segments (this is also visible, perhaps more so, in the zoom-ins of Figure 18). Given these 1D channel elements, the 2D mesh is then generated and post-processed as described — see Figures 17 and 18, where it can be noted that the 2D elements of the generated mesh are constrained along the channel centerlines, grading out to larger 2D element sizes away from the channels, all while maintaining high quality. Specifically, the generated 2D mesh has a mean element quality of $q = 0.97$, with approximately 99% of the 183,610 elements of the mesh having a quality of $q > 0.83$ and only 18 elements (corresponding to 0.01% of the elements) having a quality of $q < 0.50$, with the minimum element quality being $q = 0.33$. The mesh was generated in 13.16 minutes.

## 5.2 The Walnut Gulch Experimental Watershed

The Walnut Gulch Experimental Watershed (WGEW), established in southeastern Arizona in the 1950s, operates as an outdoor laboratory for studying hydrologic and erosion processes. Over the years, an extensive database of precipitation, runoff, and sediment records has been collected (Renard et al. (2008); Goodrich et al. (2008); Stone et al. (2008)), making it, like the





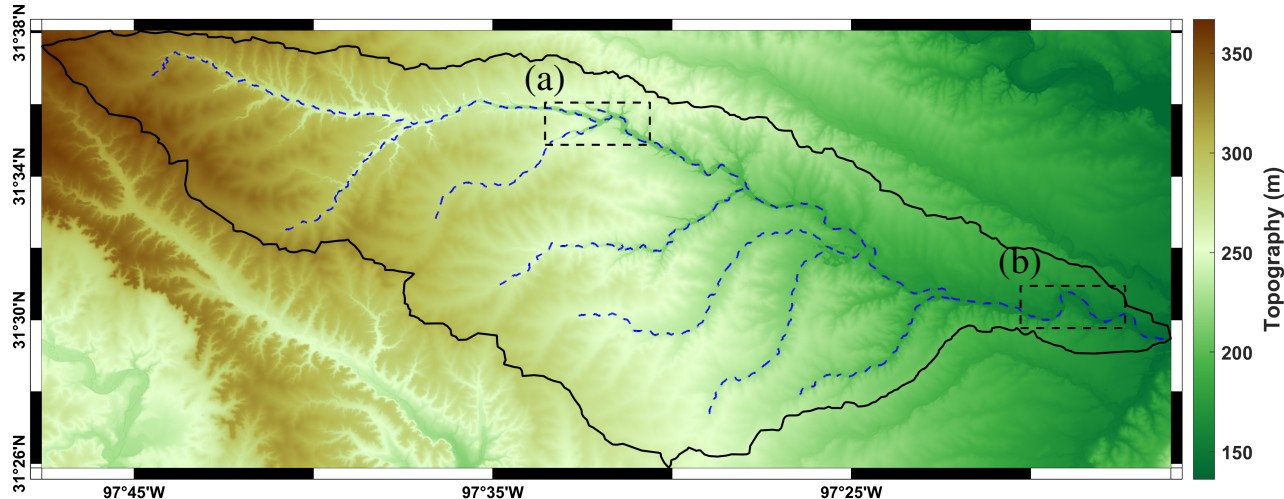

**Figure 15.** Domain of Middle Bosque River watershed (boundary of domain (*black solid line*), open-channels extracted with TopoToolbox (*blue dashed lines*), and zoom boxes (*black dashed lines*).

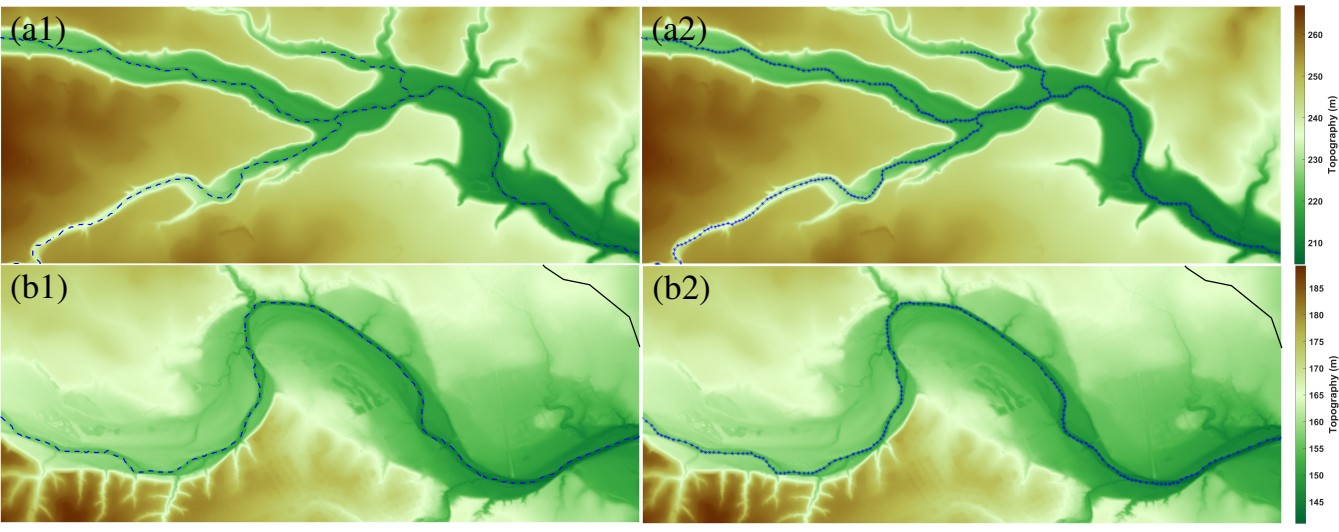

**Figure 16.** (a1,a2) Zoom-in figure on box (a) in Fig. 15 with open-channels (a1) and 1D mesh on open-channels (a2). (b1,b2) Zoom-in figure on box (b) in Fig. 15 with open-channels (b1) and 1D mesh on open-channels (b2).



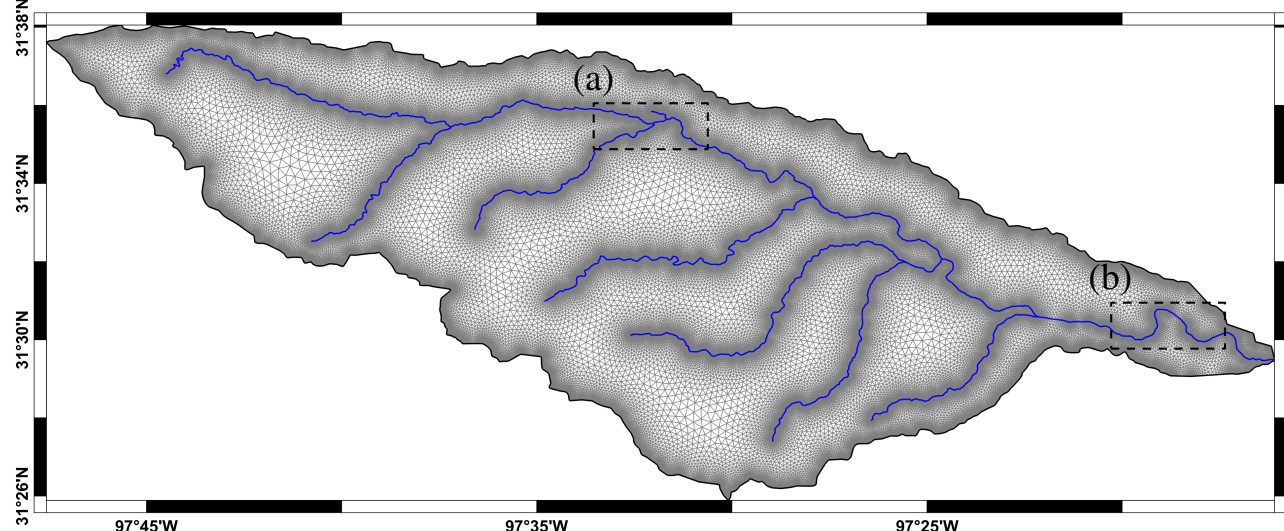

**Figure 17.** Generated mesh of the Middle Bosque River watershed, where the blue lines indicate the open channels that serve as internal constraints. Dashed rectangles labeled (a) and (b) indicate the areas of zoom-in shown in Figure 18.

previous test case, the subject of numerous studies (see, for example, Meng et al. (2008); Goodrich et al. (2012); Yu and Duan (2017)) and an ideal test case for the developed mesh generator.

The WGEW covers approximately 149 km$^2$ in Cochise county in southeastern Arizona. As with the previous test case, the boundary of the domain is obtained from USGS WBD (U.S. Geological Survey (2014)), which provides the input (polygonal) domain $\Omega_{2D}$ for the mesh generation process. Additionally, for the WGEW, both fine scale (1-m) and coarse scale (1/3-arc second) DEMs are available through the USGS 3DEP. The NHD dataset for channel centerlines is also available, but as with the previous test case, TopoToolbox is used to extract open-channels. The top panel of Figure 19 shows the domain boundary, the DEM, and the extracted channel networks of the WGEW.

Given these inputs, the mesh is generated with the following user-defined parameters: Minimum and maximum mesh sizes of 30 m and 500 m, respectively, the number of elements per radian $K$ is 20, grading limit is 0.15, and the smoothing RMSE is set to 10 m. The mesh that is generated is shown in Figures 19 and 20. Like the previous test case, the 2D elements of the generated mesh are constrained along the channel centerlines, grading out to larger 2D element sizes away from the channels, while maintaining high quality throughout the mesh. Again, as with the previous test case, the mesh has a mean element quality of $q = 0.97$. Furthermore, out of 115,459 elements, only 1,155 elements (corresponding to 1 percentile) have quality lower than $q = 0.83$ and only 12 elements (corresponding to 0.01 percentile) have quality below $q = 0.52$, with the minimum element quality being $q = 0.29$. The mesh was generated in 9.61 minutes.



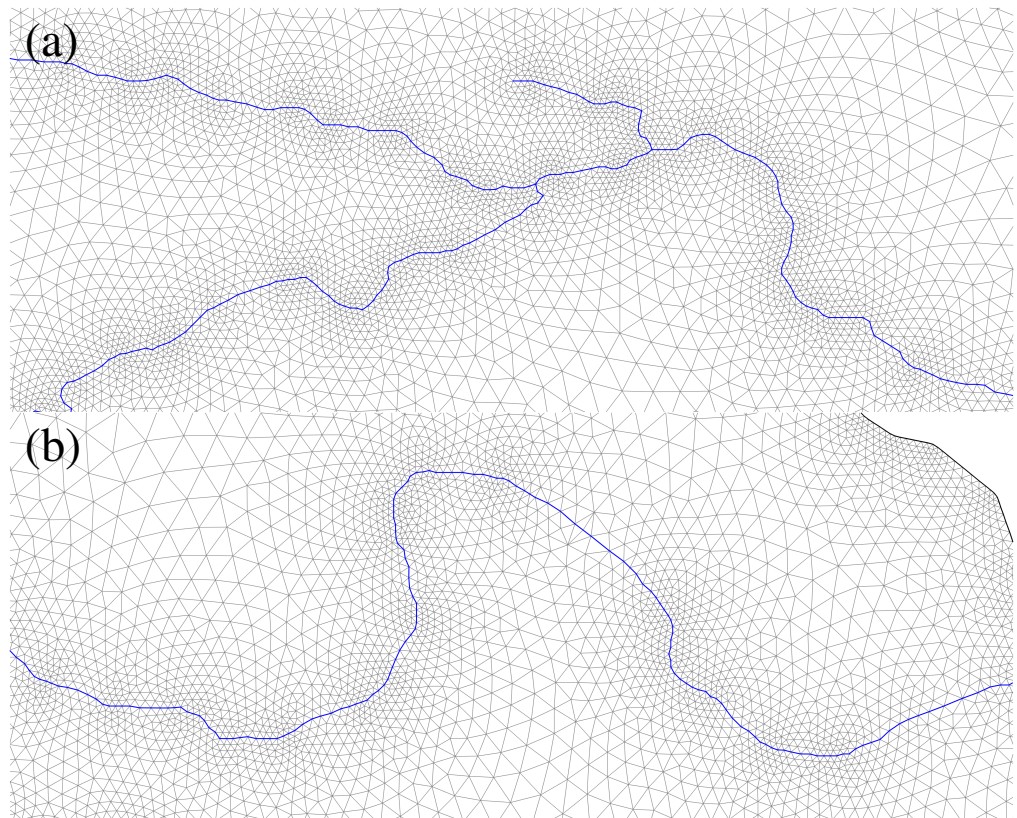

**Figure 18.** Zoom-in figures on boxes (a) and (b) of Figure 17.

### 5.3 Tidal Neches River Watershed

The Neches River flows southeast for approximately 670 km entering into Sabine Lake and then into the Gulf of Mexico near
Port Neches (see Texas Parks and Wildlife Department (1974)). This case study is focused on the Tidal Neches River segment,
which stretches approximately 45 km from the Salt Water Barrier to Sabine Lake and whose drainage area is approximately 545
km$^2$ (Schramm and Jha (2020)). This area is routinely included in the computational domains used for storm surge simulations
in Texas and southwestern Louisiana (see, for example, Dawson et al. (2011); Bunya et al. (2010)) and includes rivers and
streams of widely varying scales. For example, the Neches river, which is the main channel of the study area, has a width of
approximately 300 m, which has small tributaries with widths on the order of 10 to 30 m. The complex geometry in the study
area is not limited to the channels. Additionally, it includes a number of islands, whose areas range from 10 m$^2$ to 300 km$^2$.

For simplicity of presentation, a rectangular study area (see black line of Figure 21) is chosen as the domain boundary,
where the identification of internal constraints associated with land/water mask decomposition will be applied. The water mask
of the study area is obtained from shoreline data provided by the National Oceanic and Atmospheric Administration (NOAA)
Continually Updated Shoreline Product (CUSP) (see National Oceanic and Atmospheric Administration (NOAA) (2011)). The



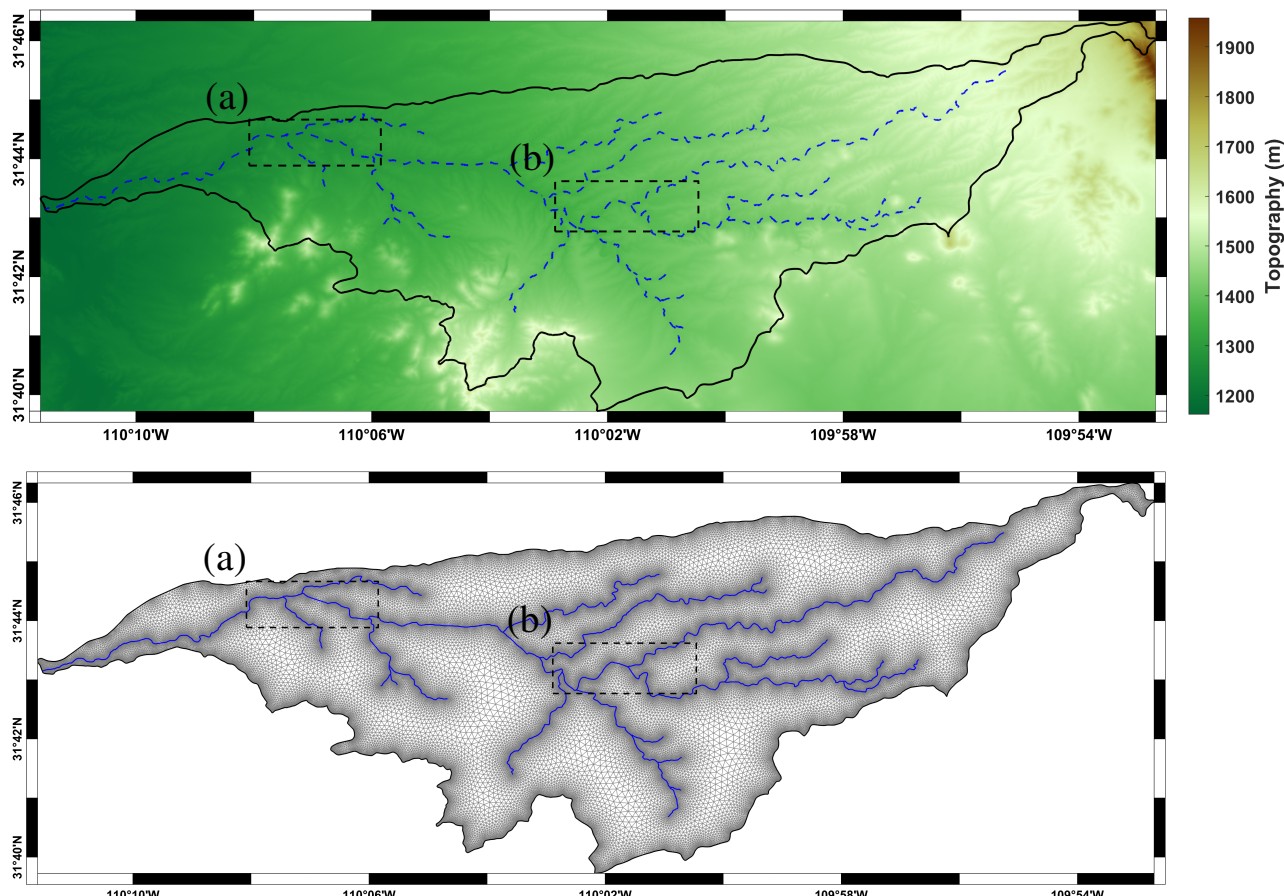

**Figure 19.** Domain of Walnut Gulch Experimental Watershed (*top*, boundary of domain (*black solid line*), open-channels extracted with TopoToolbox (*blue dashed lines*), and zoom boxes (*black dashed lines*) and 2D mesh on the study area (*bottom*).

NOAA CUSP provides a set of line segments as polylines in shapefile format (see Fig. 21). Successive line segments, which are connected to each other, are merged to construct the boundary of the water mask (see Fig. 22). The land mask is then obtained by subtracting the water mask from the rectangular study area. As with the previous two test cases, a DEM for the domain is obtained through the USGS 3DEP.

First, the water mask is pre-processed following the steps described in Section 3.2.2, with "small" islands of area less than 2,000 m² being filtered out and by using $\delta_w = 100$ m. The mesh is then generated with the following user-defined parameters: Minimum mesh size of 100 m, maximum of 1,000 m, number of elements per radian $K$ is 20, grading limit is 0.15, and smoothing RMSE is 5 m.

The 2D mesh generated from the water mask (shoreline data) has three types of internal constraints: open-channels (1D domain), internal boundaries, and boundaries between water bodies and land (see Figures 23 and 24). The algorithm automatically identified 333 open-channels, 180 internal boundaries, and 68 boundaries between water bodies and land. This identification





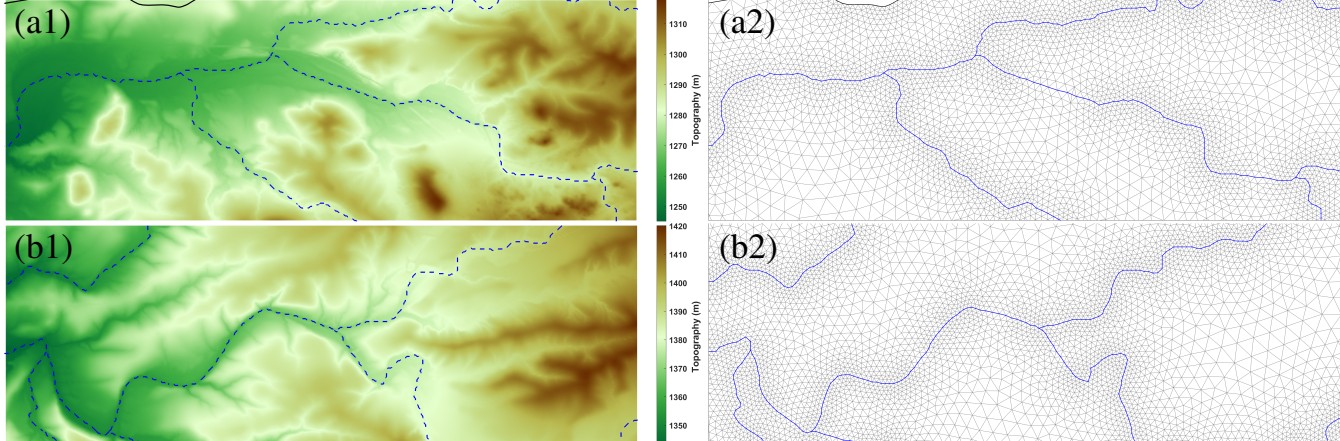

**Figure 20.** (a1,a2) Zoom-in figure on box (a) in Fig. 19 with open-channels (a1) and 2D mesh (a2). (b1,b2) Zoom-in figure on box (b) in Fig. 19 with open-channels (b1) and 2D mesh (b2).

allows the preservation of most of the channel networks (in particular, see panels (a) and (b) of Figure 24) and small-scale islands (in particular, see panel (c) of Figure 24) in the water bodies without using extremely small elements and provides a sharp delineation between land and water. It should be noted that there are a few "narrow" channels that were not identified

as 1D domains. This occurs for open-channels with free-ends, which correspond to an order 1 MA branch, as a result of MA pruning (see Step 2 in Section 3.2.1). Likewise, there are a few small-scale islands that are not identified as internal boundaries. However, overall, the algorithm does an exceptional job of automatically identifying the internal constraints based on the specified width parameter, while maintaining elements of high quality. The 2D mesh has a mean element quality of $q = 0.93$, with only 624 (out of 62,403) elements having element quality lower than $q = 0.65$ (corresponding to 1 percentile) and only

6 elements (corresponding to 0.01 percentile) having element quality lower than $q = 0.25$. The minimum element quality in this case is $q = 0.16$, which is lower than the previous two test cases. This is a result of internal constraints being close to one another and is a trade-off for preserving geographic features.

## 6 Conclusions

An automatic mesh generation algorithm with internal constraints, especially for coupled 1D/2D hydrodynamic models, is

presented in this paper. The main objectives of the proposed algorithm are to automatically identify internal constraints (mainly channel centerlines) in the domain and to generate collocated meshes along the internal constraints with efficient sizing. The identification of internal constraints is developed for both land and water subdomains. TopoToolbox is used to extract channel centerlines from land subdomains, and an additional smoothing is applied to estimate appropriate curvature of the lines. The extraction of internal constraints from water subdomains is based on a width function and a user-defined threshold of thin

channels. This enables identification of three types of internal constraints if water mask is given. Several additional processes



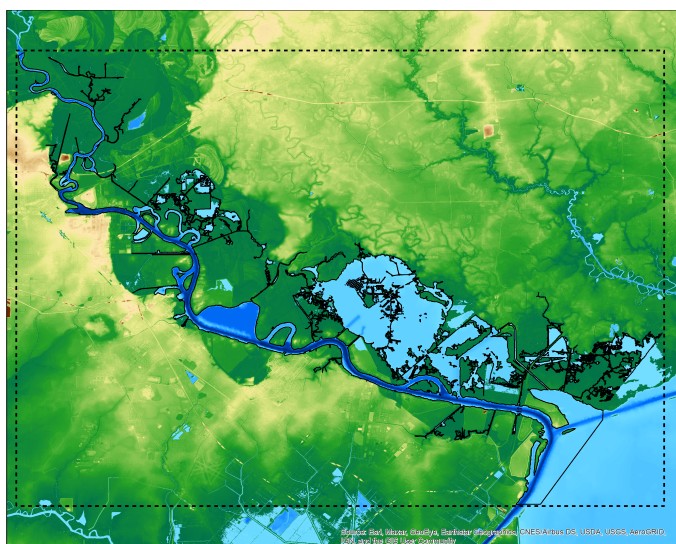

**Figure 21.** Domain of Lower Neches Basin. Boundary of study area (*black dashed* lines) and shorelines provided by NOAA CUSP (*black solid* lines).

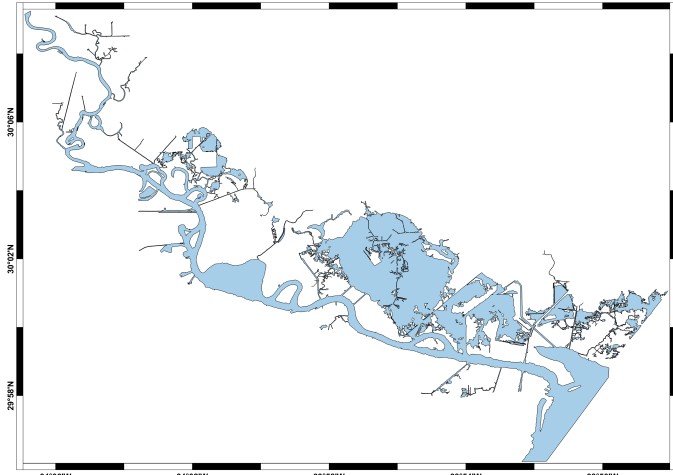

**Figure 22.** Water mask of Lower Neches Basin obtained from shoreline data provided by NOAA CUSP (blue shaded areas). The land mask is obtained by subtracting the water mask from the rectangular study area.



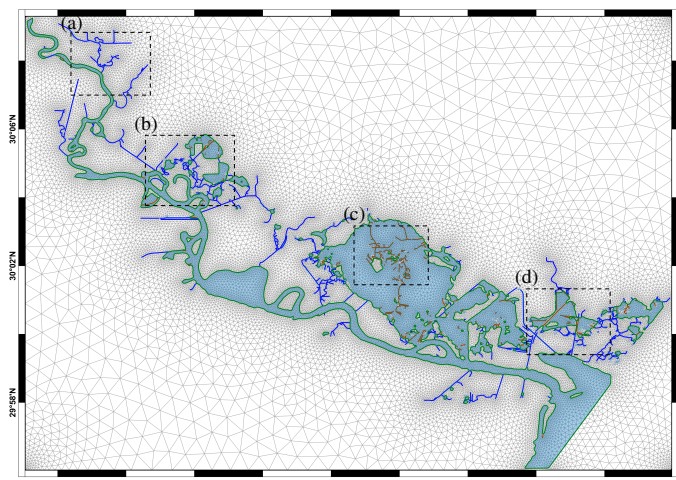

**Figure 23.** 2D mesh on Lower Neches Basin with three types of internal constraints (channel centerlines (*blue*), internal boundaries (*brown*), and shoreline (*green*)).

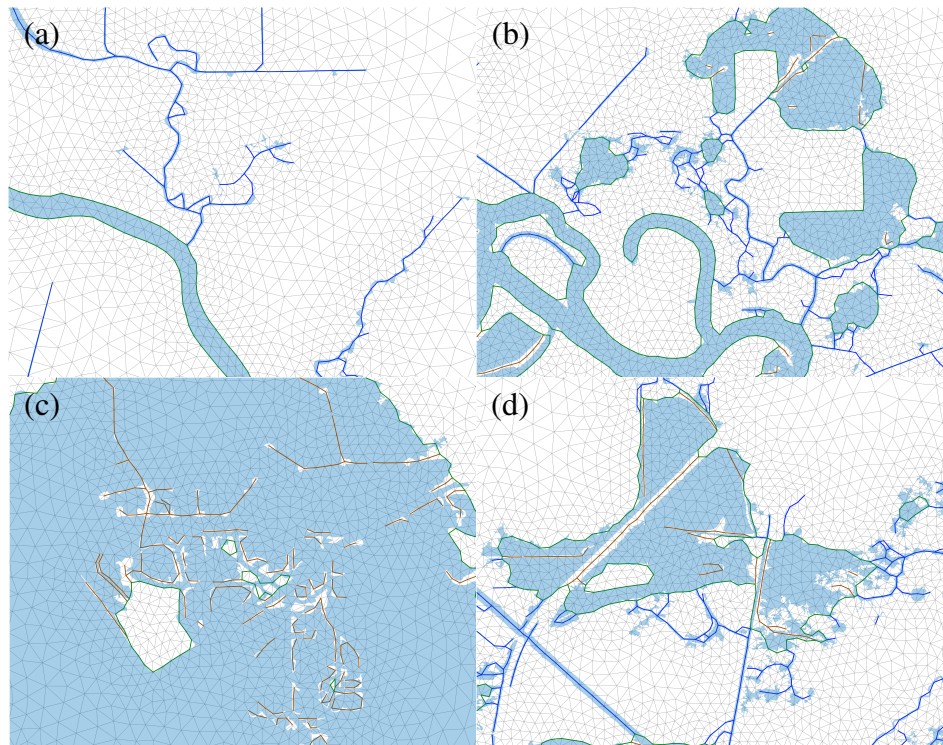

**Figure 24.** Zoom-in figures on corresponding boxes of Figure 23, where again channel centerlines are shown in *blue*, internal boundaries in *brown*, and shorelines in *green*.





are developed for complex water subdomains including representation of thin islands as internal boundaries. The meshes generated with the proposed algorithm have precise alignment along the given internal constraints with efficient sizing of high-quality 2D elements. This is obtained by assigning proper target mesh sizes to the 1D/2D force equilibrium algorithms and applying post-processing of 1D elements and density control.

While the test cases presented in this paper have, in general, elements of high quality, there are still a few elements of poor quality. This can occur when internal constraints are located too close to each other. For example, thin channels that are very close to each other or thin islands that are located very close to a shoreline. Note that these cases can be resolved by ignoring the object or allowing very small element sizes if the element quality is of higher priority. However, the proposed algorithm places a higher priority on keeping geographical features with relatively low resolution.

Future work may include the following two objectives. First, an efficient background grid such as an octree or unstructured grid can be used to improve the computational efficiency, especially for the identification of internal constraints in water subdomains. A key factor of the identification of "thin" regions is the computation of the width function, which requires that the background grid is fine enough to span thin regions. It is expected that use of an efficient background grid such as an octree or unstructured grid can improve the computational efficiency. Second, an automatized algorithm to retrieve cross-sectional

profiles from channels would be beneficial. Note that channel cross-sectional representations, which are typically specified as triangles, rectangles, or trapezoids, are required for most of coupled 1D/2D hydrodynamic models. While the cross-sections of channels in land subdomains can be detected from the DEM, there is some ambiguity for the width of the channels. On the other hand, the width of channels in the water subdomain is relatively clear as it can be identified with the water mask. However, the cross-sectional information would need to be provided by supplemental bathymetric survey data, as "standard"

DEMs do not contain bathymetric elevations.

**Appendix A: Computation of medial axis with vector distance transform**

In this section, we provide details of the medial axis computation briefly described in Section 3.2.1 and Eq. 8. In particular, we demonstrate the fact that the divergence of the VDT has positive values only on the medial axis. This is based on Voronoi polygons and their properties described in Lee (1982). A given polygon $P$ can be partitioned into a set of Voronoi polygons,

with boundaries referred to as Voronoi edges (see Fig. A2 for example). Voronoi polygons can be categorized into two types. One consists of Voronoi polygons whose boundaries include a segment of the external boundary $\partial P$. Another consists of Voronoi polygons whose boundaries do not include any segments of the external boundary. We refer to these two types as *lateral* and *wedge* type, respectively (the white and gray polygons, respectively, in Fig. A2).

    In the case of lateral type Voronoi polygons, the terminal points of the VDT are on the corresponding external boundary

segment (see Lemma 1 in Lee (1982) and Fig. A2). Note that by the definition of VDT, the VDT is perpendicular with the corresponding external boundary segment. This VDT can be represented as, once the corresponding external boundary segment is projected to $y = 0$ (see Fig. A1),

$$\mathbf{V}(\mathbf{x}, P) = (0, -y) \text{ where } \mathbf{x} = (x, y). \tag{A1}$$



In the wedge type Voronoi polygons, the terminal points of the VDT are vertices of a given polygon (see Lemma 1 in Lee

(1982) and Fig. A2). This VDT can be represented as, once the corresponding vertex is projected to the origin (see Fig. A1),

$$\mathbf{V}(\mathbf{x}, P) = (-x, -y), \text{ where } \mathbf{x} = (x, y). \tag{A2}$$

Therefore, we have

$$\nabla \cdot \mathbf{V}(\mathbf{x}, P) = \begin{cases} -1 & \text{in lateral type Voronoi polygons} \\ -2 & \text{in wedge type Voronoi polygons} \end{cases}. \tag{A3}$$

Now, $\nabla \cdot \mathbf{V}$ is computed on the Voronoi edges. From Corollary 3 in Lee (1982), the medial axis is a subset of the Voronoi

edges. Let us call the Voronoi edges that are not part of the medial axis as *extra* Voronoi edges. Note that, by Corollay 3 in

Lee (1982), the extra Voronoi edges are a subset of the boundaries between lateral and wedge type Voronoi polygons. With the

projection of external boundaries and vertices described above and Eq. A1 and A2, it can be shown that VDT is continuous but

non-differentiable on extra Voronoi edges. Therefore, the divergence of VDT cannot be analytically defined on extra Voronoi

edges, but here we show that numerical divergence of VDT is between $-1$ and $-2$ on extra Voronoi edges. For simplicity, let

us assume that the extra Voronoi edge is projected to $x = 0$, which is the case shown in Fig. A1. Since the VDT is continuous

and differentiable on each lateral and wedge type Voronoi polygons, forward and backward difference schemes on $\mathbf{x} = (0, y)$

give $\nabla \cdot \mathbf{V}(\mathbf{x}, P) = -2$ and $-1$, respectively. A central difference scheme on $\mathbf{x} = (0, y)$ gives

$$\nabla \cdot \mathbf{V}(\mathbf{x}, P) = \lim_{\Delta x \to 0} \frac{-\Delta x - 0}{2\Delta x} + \lim_{\Delta y \to 0} \frac{-(y + \Delta y) + (y - \Delta y)}{2\Delta y} \tag{A4}$$

$$= -\frac{1}{2} + \frac{-2\Delta y}{2\Delta y} \tag{A5}$$

$$= -1.5. \tag{A6}$$

Therefore, the numerical divergence from forward, backward, and central difference schemes is between $-1$ and $-2$ on ex-

tra Voronoi edges. Finally, on the medial axis, the VDT is discontinuous and the divergence cannot be computed explicitly.

However, given the fact that the VDT diverges from the medial axis to external boundaries, the numerical divergence must be

positive. The $\nabla \cdot \mathbf{V}$ is numerically computed with `divergence` function in MATLAB and shown in Fig. A3.

One advantage of this method is that it requires low additional computational cost. In our mesh generation algorithm, the

VDT is computed as part of computing the distance map (note that $|d(\mathbf{x}, \partial P)| = \|\mathbf{V}(\mathbf{x}, P)\|$), which is as essential requirement.

Therefore, additional step is simply computing divergence of VDT. The MA can also be found with criteria based on the

gradient of the distance map $d(\mathbf{x}, \partial P)$ (see Koko (2015); Roberts et al. (2019)), .i.e.,

$$\|\nabla d(\mathbf{x}, \partial P)\| < \epsilon < 1, \tag{A7}$$

where $\epsilon$ is a user-specified parameter (typically taken to be 0.9); however, based on our experiments, the MA computed from

Eq. 8 tends to be more accurate than the MA computed from Eq. A7.



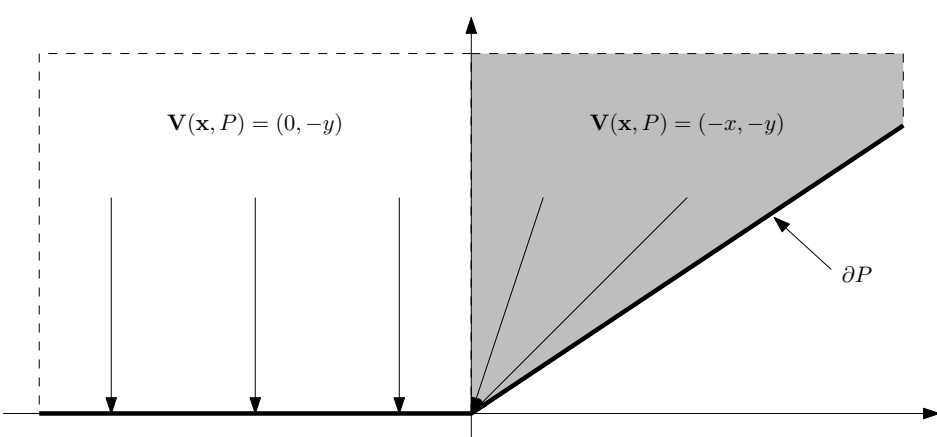

**Figure A1.** Schematic of VDT in lateral and wedge type Voronoi polygons (*white* and *gray* polygons, respectively).



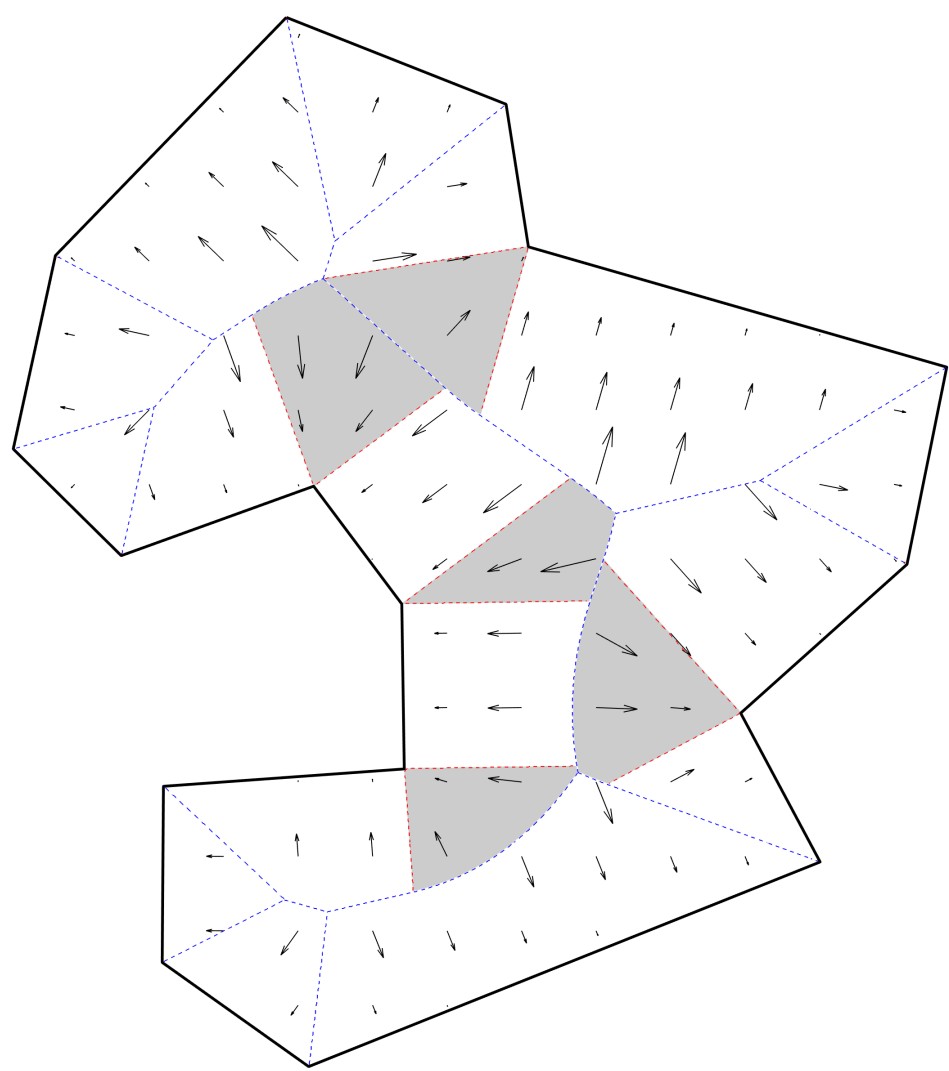

**Figure A2.** Voronoi polygons of a simple polygon, where boundaries of Voronoi polygons are shown as dashed lines (*blue* lines are the medial axis (a subset of the Voronoi edges) and *red* dashed lines are Voronoi edges which are not part of the medial axis). The vector distance transform is shown by black arrows (scaled for visualization purpose). *White* and *gray* polygons indicate *lateral* and *wedge* type Voronoi polygons, respectively.



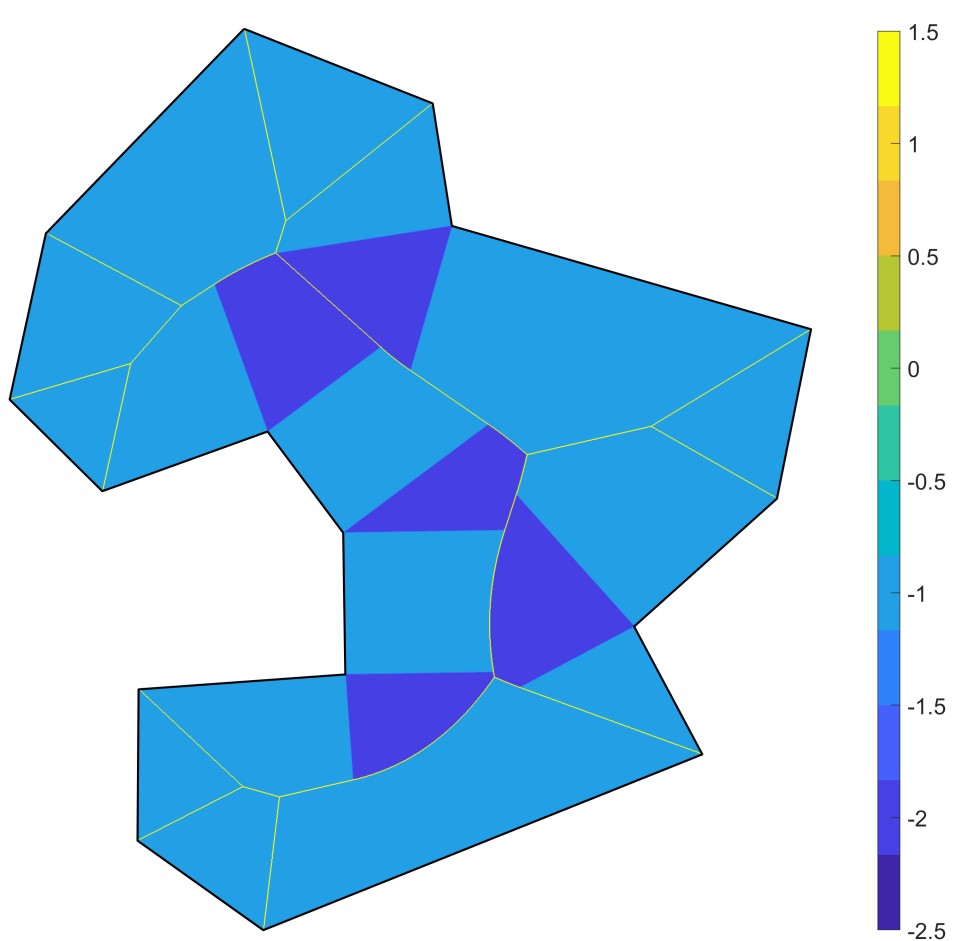

**Figure A3.** Divergence of the vector distance transform of a simple polygon. Note that the divergence values are -1 and -2 for lateral and wedge type Voronoi polygons and positive on the medial axis.





*Code availability.* The current version of ADMESH+, the mesh generator presented in this study, is available on the following GitHub page: https://github.com/younghun-kang/ADMESH. The ADMESH+ is under active development and the latest version is available upon request.

*Author contributions.* EK made an initial suggestion and guided this work. YK and EK equally contributed to the development of the
framework. YK developed the model code and performed it for the given test cases.

*Competing interests.* The authors declare that they have no conflict of interest.

*Acknowledgements.* The authors would like to acknowledge the support of National Science Foundation grant ICER-1854991.



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
