# Peer review of "An automatic mesh generator for coupled 1D/2D hydrodynamic models"

_EGUsphere, 2023_

## Author Response (AR1)

**Authors response to the open discussion on "An automatic mesh generator for coupled 1D/2D hydrodynamic models.": EGUSPHERE-2023-1434**

Referees comments (RC1 and RC2) are in black, and point-by-point response by authors (AC) are in blue.

**Response to referee #1**

RC1: Line 14 – Drop "as" .... routinely used to simulate....
AC: Thank you for pointing out the grammatical error. It has been fixed.

RC1: Lines 21 and 22 – Provide a citation justifying or showing an example support for your statement about adequately described by simpler one-dimensional....
AC: We appreciate pointing out the need for citations. We added some citations and the line has been slightly edited for clarification.

RC1: Line 67 – add "an" ... models that is an extension....
AC: Thank you for pointing out the grammatical error. It has been fixed.

RC1: Line 79... should also mention that an element size function is often supplied
AC: We kindly note that the element size function is mentioned in the previous paragraph.

RC1: Lines 156-157 – this section should include a longer discussion on the importance of the quality of a DEM and a discussion on the sensitivity of the quality of the DEM to the results of ADMESH+.
AC: We appreciate your suggestion. In response, we added a longer discussion on the quality of DEM and the extracted open-channels.

RC1: Line 273 – add "to" ...constraints too close to each other.
AC: Thank you for pointing out the grammatical error. It has been fixed.

RC1: Line 310 – Provide more details on how exactly ordering of the medial axis branches is done.
AC: We kindly note that the details are described in Section 3.2.1, Step 2.

RC1: Lines 320-326 — A longer discussion is needed on the sensitivity of the ADMESH+ algorithms to user setting/DEM quality for level 1 and level 2 regions and mis-identification of narrow features and the the omission of some level 1 land regions. How can one check/find when this is happening and what are solutions for correcting it when it does happen?
AC: We appreciate pointing out the need for a longer discussion of this topic. We have added potential solutions for the mis-classified level 1 regions.

RC1: Line 339 – Provide more explanation for the use of curvature of internal constraints for mesh size and why it is a good choice.
AC: When we approximate internal constraints with linear elements, the errors will be higher for curved parts and lower for flat parts. It results in the need of small sized elements for curved parts. The use of

curvature of internal constraints allows assigning a mesh size function such that its target mesh size is higher for curved parts and lower for flat parts.

RC1: Line 360 – The choice of 1 to 10 meters needs more explanation. Is it tied to the resolution and quality of the DEM and/or to the desired minimum element size?
AC: As we discussed in the manuscript, the choice of 1 to 10 meters is based on our numerical experiments. It may relate to the DEM resolution rather than minimum element size, and it has been noted in the manuscript.

RC1: Line 374 – Remove second "." after Eq. 33.
AC: Thank you for pointing out the grammatical error. It has been fixed.

RC1: Line 411 – A question here? After removing 1D elements, does this require another round of adjustments to balance out the spacing?
AC: By "removing 1D elements", we meant merging 1D elements to their neighboring elements. The manuscript has been edited to clarify this. Also, note that it may be ideal to adaptively merge such elements (based on the lengths of left and right elements), but they are simply merged to left elements in the current version.

RC1: Line 420 – What is the distance (hmin/2) in relation to? The node's distance to the internal constraints? Suggest making this more clear.
AC: It's distance to the internal constraints, and the manuscript has been edited to make it clearer.

RC1: Line 430 – Explain why this particular mesh element quality measure was selected. Were other quality measures examined?
AC: A longer discussion of the mesh element quality has been added in the manuscript. However, other element qualities have not been examined. We may add it for our future work.

RC1: Line 482 – The minimum mesh quality of 0.29 seems very low and would possibly lead to numerical model instabilities and or accuracy loss. Do the authors have a suggested minimum mesh quality to allow and if so, what should be done to fix elements that fall below that threshold?
AC: Based on our previous work (Conroy et al. (2012) doi.org/10.1007/s10236-012-0574-0), we suggest minimum element quality 0.30 without internal constraints. However, we have not yet established a suggestible minimum element quality with internal constraints. An additional discussion for possible solution for the poor quality elements has been added in the conclusion section.

RC1: Figure 18(b) shows that there are some nodes that have a high valence count, i.e. are connected to a large number of surrounding nodes (optimal would be 6 connected nodes). This high valance typically happens when element size transitions happen too rapidly. What is causing this in your algorithm? Are there catches/fixes that could be implemented during the overall process?
AC: The high valance may be caused by the internal constraints; however, we may need additional examinations for a more accurate answer for this question. An additional discussion for possible solution for the poor quality elements has been added in the conclusion section.

RC1: Lines 496-497 – How are the successive line segments merged? What tool is used?
AC: The modified shoreline data are provided by our colleague and we believe they are merged manually. For the exact information of which software has been used, we will need to reach out to our colleague.

RC1: Lines 502 and 503 – Discuss why K=20, grading limit 0.15, and RSME of 5 meters were selected and what impacts altering those values would have to the mesh construction.
AC: These parameters are chosen as it gives "reasonable" results in our numerical experiments as shown in our previous work (see, Conroy2012) and this work.

RC1: Lines 516-517 ... Please add a zoomed in image showing an example area where the internal constraints are too close to one another and resulted in a low element quality. Also, discuss what a user would need to

do in order to fix these issues.

AC: As suggested by the reviewer, we have added zoomed-in figures. As discussed in the conclusion section, it can be resolved by ignoring the object or allowing very small element sizes if the element quality is of higher priority.

RC1: Line 582 – The sentence "Therefore, additional step is .... of VDT." does not seem to be well formed and is out of context. Maybe remove it, or add clarification.

AC: Thank you for pointing out the need for clarification. The manuscript has been edited to clarify it.

RC1: Line 587 – Suggestion adding if not already provided, that the sample generated meshes given in this paper also be shared alongside of the source code.

AC: We appreciate the great suggestion. The sample meshes have been provided alongside the source code.

**Response to referee #2**

RC2: P1-L15: It is not obvious how the sea-level rise itself is assessed using 2D hydrodynamic models. If this is meant to be considering the effect of sea level rise to coastal flooding, it may be better to rephrase it or take a different example.
AC: Thank you for pointing out this is not an obvious example. We have taken another example.

RC2: P1-L20: "expensive" `-->` "expense"
AC: Thank you for pointing out the grammatical error. It has been fixed.

RC2: P2-L60: A more recent work that is not mentioned here for internally-connected domains has been published: Bunya, et al. (2023) https://doi.org/10.1016/j.advengsoft.2023.103516.
AC: Thank you for pointing out that the good reference has been missed. It has been added in the previous paragraph (P2-L45) than the reviewer mentioned, as we believe it would be a better place to add this citation.

RC2: P3-L91: "Section" `-->` "section"
AC: Thank you for pointing out the grammar error. It has been fixed.

RC2: P4-L105: It is not clear which work the authors refer to as "our previous work".
AC: Thank you for pointing out the unclarity. We have added the citation of the work.

RC2: P6-L143: It would be helpful for readers if the definition of the global flow matrix M is presented in a brief manner.
AC: We appreciate the suggestion for better clarity. The definition has been added as suggested.

RC2: P8-L184: The definition of "sharp" corners doesn't appear to be presented.
AC: Thank you for pointing this out. The sharpness is related to the threshold parameter $\delta_\theta$. However, we decided to remove the word "sharp" as it may cause confusion.

RC2: P8-L190: Some more description of "background grid points" should be added here or somewhere earlier. Otherwise, it's not clear how they are defined.
AC: We kindly note that the background grid points are described in Section 2.

RC2: P13-L252 and L253: Can these equations be expressed using symbols other than the leftwards arrows?
AC: As suggested, the symbol has been replaced with colon equal. Furthermore, for better clarity, the notations of masks are modified.

RC2: P13-L264: Present the definitions of "Perimter" (typo?) and "Area".
AC: Thank you for pointing out the typo. It has been fixed. Also, the "perimeter" and "area" are standard measure. In order to avoid confusion, they are replaced to lower cases.

RC2: P15-L299: "the updated level 1 mask" `<--` Is this land mask or water mask?
AC: It is the level 1 mask from the updated water mask. It has been clarified in the manuscript. Also, as responded to the earlier comment, the notations of masks are modified for better clarity.

RC2: P16 or somewhere earlier: How are the medial axes points connected to construct line segments?
AC: We appreciate pointing out that the description of the construction of medial axis branches are missed in the manuscript. The description has been added in the appendix.

RC2: P17 or somewhere earlier: The force equilibrium algorithm used in this work can be briefly presented as it plays a central role in Section 4.
AC: Thank you for the suggestion. A brief description of the force equilibrium algorithm has been added in the manuscript.

RC2: P19-L393: "... there should be at least three 1D elements along the boundaries of the level 2 regions. ..." **<---** This may require some more elaboration for clarity.
AC: Thank you for pointing out the unclarity. It has been edited for clarity.

RC2: P19-L394: "... identified, as level 2, regions, ..." **<---** Check if the commas are necessary.
AC: Thank you for pointing out the grammatical error. It has been fixed.

RC2: P20-L404: "... then the 1D meshes are removed." **<---** Are both 1D meshes removed? Or, do you leave one of them?
AC: 1D meshes for open-channel/internal boundaries are removed. For better clarity, it has been edited.

RC2: For three test cases in Section 5: The minimum and maximum element sizes of the resulting meshes should be presented to show how they coincide with or differ from the specified values.
AC: As suggested, the minimum and maximum element sizes have been presented for the test cases.